# `TACO`: Temporal Latent Action-Driven Contrastive Loss for Visual Reinforcement Learning

**Ruijie Zheng**[1]     **Xiyao Wang**[1]
**Yanchao Sun**[1]     **Shuang Ma**[4]     **Jieyu Zhao**[1,2]
**Huazhe Xu**[3,4,§]     **Hal Daumé III**[1,5,§]     **Furong Huang**[1,§]

[1] University of Maryland, College Park [2] University of Southern California
[3] Tsinghua University [4] Shanghai Qi Zhi Institute
[5] Microsoft Research
rzheng12@umd.edu

## Abstract

Despite recent progress in reinforcement learning (RL) from raw pixel data, sample inefficiency continues to present a substantial obstacle. Prior works have attempted to address this challenge by creating self-supervised auxiliary tasks, aiming to enrich the agent's learned representations with control-relevant information for future state prediction. However, these objectives are often insufficient to learn representations that can represent the optimal policy or value function, and they often consider tasks with small, abstract discrete action spaces and thus overlook the importance of action representation learning in continuous control. In this paper, we introduce `TACO`: **T**emporal **A**ction-driven **CO**ntrastive Learning, a simple yet powerful temporal contrastive learning approach that facilitates the concurrent acquisition of latent state and action representations for agents. `TACO` simultaneously learns a state and an action representation by optimizing the mutual information between representations of current states paired with action sequences and representations of the corresponding future states. Theoretically, `TACO` can be shown to learn state and action representations that encompass sufficient information for control, thereby improving sample efficiency. For online RL, `TACO` achieves 40% performance boost after one million environment interaction steps on average across nine challenging visual continuous control tasks from Deepmind Control Suite. In addition, we show that `TACO` can also serve as a plug-and-play module adding to existing offline visual RL methods to establish the new state-of-the-art performance for offline visual RL across offline datasets with varying quality.

## 1 Introduction

Developing reinforcement learning (RL) agents that can perform complex continuous control tasks from high-dimensional observations, such as pixels, has been a longstanding challenge. A central aspect of this challenge lies in addressing the sample inefficiency problem in visual RL. Despite significant progress in recent years [33, 34, 53, 52, 16, 15, 17, 18, 21, 48], there remains a considerable gap in sample efficiency between RL with physical state-based features and those with pixel inputs. This disparity is particularly pronounced in complex tasks; thus tackling it is crucial for advancing visual RL algorithms and enabling their practical application in real-world scenarios.

---

[§]Equal advising

[*]Our code is released at https://github.com/FrankZheng2022/TACO.

37th Conference on Neural Information Processing Systems (NeurIPS 2023).

State representation learning has become an essential aspect of RL research, aiming to improve sample efficiency and enhance agent performance. Initial advancements like CURL [33] utilized a self-supervised contrastive InfoNCE objective [47] for state representation, yet it overlooks the temporal dynamics of the environment. Subsequent works, including CPC [23], ST-DIM [2], and ATC [44], made progress to rectify this by integrating temporal elements into the contrastive loss, linking pairs of observations with short temporal intervals. The objective here was to develop a state representation capable of effectively predicting future observations. A more comprehensive approach was taken by DRIML [37], which incorporated the first action of the action sequence into the temporal contrastive learning framework. However, these methods, while innovative, have their shortcomings. The positive relations in their

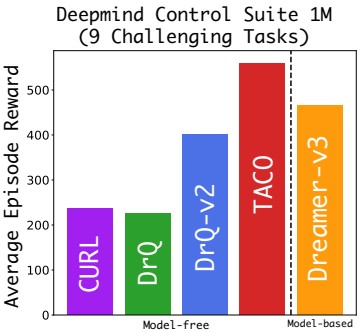

**Figure 1:** Comparison of average episode reward across nine challenging tasks in Deepmind Control Suite after one million environment steps.

contrastive loss designs are often policy-dependent, potentially leading to instability during policy updates throughout the training process. Consequently, they lack the theoretical foundation needed to capture all information representing the optimal policy. Furthermore, these methods except for CURL and ATC typically focus on environments such as Atari games with well-represented, abstract discrete action spaces, thereby overlooking the importance of action representation in continuous control tasks [1, 28]. However, by learning an action representation that groups semantically similar actions together in the latent action space, the agent can better generalize its knowledge across various state-action pairs, enhancing the sample efficiency of RL algorithms. Therefore, learning both state and action representations is crucial for enabling the agent to more effectively reason about its actions' long-term outcomes for continuous control tasks.

In this paper, we introduce **T**emporal **A**ction-driven **CO**ntrastive Learning (`TACO`) as a promising approach to visual continuous control tasks. `TACO` simultaneously learns a state and action representation by optimizing the mutual information between representations of current states paired with action sequences and representations of the corresponding future states. By optimizing the mutual information between state and action representations, `TACO` can be theoretically shown to capture the essential information to represent the optimal value function. In contrast to approaches such as DeepMDP [12] and SPR [42], which directly model the latent environment dynamics, our method transforms the representation learning objective into a self-supervised InfoNCE objective. This leads to more stable optimization and requires minimal hyperparameter tuning efforts. Consequently, `TACO` yields expressive and concise state-action representations that are better suited for high-dimensional continuous control tasks.

We demonstrate the effectiveness of representation learning by `TACO` through extensive experiments on the DeepMind Control Suite (DMC) in both online and offline RL settings. `TACO` is a flexible plug-and-play module that could be combined with any existing RL algorithm. In the online RL setting, combined with the strong baseline DrQ-v2[52], `TACO` significantly outperforms the SOTA model-free visual RL algorithms, and it even surpasses the strongest model-based visual RL baselines such as Dreamer-v3 [18]. As shown in Figure 1, across nine challenging visual continuous control tasks from DMC, `TACO` achieves a 40% performance boost after one million environment interaction steps on average. For offline RL, `TACO` can be combined with existing strong offline RL methods to further improve performance. When combined with TD3+BC [11] and CQL [31], `TACO` outperforms the strongest baselines across offline datasets with varying quality.

We list our contributions as follows:

1. We present `TACO`, a simple yet effective temporal contrastive learning framework that simultaneously learns state and action representations.

2. The framework of `TACO` is flexible and could be integrated into both online and offline visual RL algorithms with minimal changes to the architecture and hyperparameter tuning efforts.

3. We theoretically show that the objectives of `TACO` is sufficient to capture the essential information in state and action representation for control.

4. Empirically, we show that `TACO` outperforms prior state-of-the-art model-free RL by 1.4x on nine challenging tasks in Deepmind Control Suite. Applying TACO to offline RL with SOTA

algorithms also achieves significant performance gain in 4 selected challenging tasks with pre-collected offline datasets of various quality.

## 2   Preliminaries

### 2.1   Visual reinforcement learning

Let $\mathcal{M} = \langle \mathcal{S}, \mathcal{A}, \mathcal{P}, \mathcal{R}, \gamma \rangle$ be a Markov Decision Process (MDP). Here, $\mathcal{S}$ is the state space, and $\mathcal{A}$ is the action space. The state transition kernel is denoted by $\mathcal{P} : \mathcal{S} \times \mathcal{A} \rightarrow \Delta(\mathcal{S})$, where $\Delta(\mathcal{S})$ is a distribution over the state space. $\mathcal{R} : \mathcal{S} \times \mathcal{A} \rightarrow \mathbb{R}$ is the reward function. The objective of the Reinforcement Learning (RL) algorithm is the identification of an optimal policy $\pi^* : \mathcal{S} \rightarrow \Delta(\mathcal{A})$ that maximizes the expected value $\mathbb{E}_\pi [\sum_{t=0}^\infty \gamma^t r_t]$. Additionally, we can define the optimal Q function as follows: $Q^*(s, a) = E_{\pi^*} \left[ \sum_{t=0}^\infty \gamma^t r(s_t, a_t) | s_0 = s, a_0 = a \right]$, such that the relationship between the optimal Q function and optimal policy is $\pi(s) = \arg \max_a Q^*(s, a)$. In in the domain of visual RL, high-dimensional image data are given as state observations, so the simultaneous learning of both representation and control policy becomes the main challenge. This challenge is exacerbated when the environment interactions are limited and the reward signal is sparse.

### 2.2   Contrastive learning and the InfoNCE objective

Contrastive Learning, a representation learning approach, imposes similarity constraints on representations, grouping similar/positive pairs and distancing dissimilar/negative ones within the representation space. Contrastive learning objective is often formulated through InfoNCE loss [47] to maximize the mutual information between representations of positive pairs by training a classifier. In particular, let $X, Y$ be two random variables. Given an instance $x \sim p(x)$, and a corresponding positive sample $y^+ \sim p(y|x)$ as well as a collection of $Y = \{y_1, ..., y_{N-1}\}$ of $N - 1$ random samples from the marginal distribution $p(y)$, the InfoNCE loss is defined as

$$\mathcal{L}_N = \mathbb{E}_x \left[ \log \frac{f(x, y^+)}{\sum_{y \in Y \cup \{y^+\}} f(x, y)} \right] \tag{1}$$

Optimizing this loss will result in $f(x, y) \propto \dfrac{p(y|x)}{p(y)}$ and one can show that InfoNCE loss upper bounds the mutual information, $\mathcal{L}_N \geq \log(N) - \mathcal{I}(X, Y)$.

## 3   `TACO`: temporal action-driven contrastive Loss

`TACO` is a flexible temporal contrastive framework that could be easily combined with any existing RL algorithms by interleaving RL updates with its temporal contrastive loss update. In this section, we first presnt the overall learning objective and theoretical analysis of `TACO`. Then we provide the architectural design of `TACO` in detail.

### 3.1   Temporal contrastive learning objectives and analysis

In the following, we present the learning objectives of `TACO`. The guiding principle of our method is to learn state and action representations that capture the essential information about the environment's dynamics sufficient for learning the optimal policy. This allows the agent to develop a concise and expressive understanding of both its current state and the potential effects of its actions, thereby enhancing sample efficiency and generalization capabilities.

Let $S_t$, $A_t$ be the state and action variables at timestep $t$, $Z_t = \phi(S_t)$, $U_t = \psi(A_t)$ be their corresponding representations. Then, our method aims to maximize the mutual information between representations of current states paired with action sequences and representations of the corresponding future states:

$$\mathbb{J}_{\texttt{TACO}} = \mathcal{I}(Z_{t+K}; [Z_t, U_t, ..., U_{t+K-1}]) \tag{2}$$

Here, $K \geq 1$ is a fixed hyperparameter for the prediction horizon. In practice, we estimate the lower bound of the mutual information by the InfoNCE loss, with details of our practical implementation described in §3.2.

We introduce the following theorem extending from Rakely et al. [41] to demonstrate the sufficiency of `TACO` objective:

**Theorem 3.1.** *Let $K \in \mathbb{N}^+$, and $\mathbb{J}_{TACO} = \mathcal{I}(Z_{t+K}; [Z_t, U_t, ..., U_{t+K-1}])$. If for a given state and action representation $\phi_Z, \psi_U$, $\mathbb{J}_{TACO}$ is maximized, then for arbitrary state-action pairs $(s_1, a_1), (s_2, a_2)$ such that $\phi(s_1) = \phi(s_2), \psi(a_1) = \psi(a_2)$, it holds that $Q^*(s_1, a_1) = Q^*(s_2, a_2)$.*

This theorem guarantees that if our mutual information objective Equation (2) is maximized, then for any two state-action pairs $(s_1, a_1)$ and $(s_2, a_2)$ with equivalent state and action representations, their optimal action-value functions, $Q^*(s_1, a_1)$ and $Q^*(s_2, a_2)$, will be equal. In other words, maximizing this mutual information objective ensures that the learned representations are sufficient for making optimal decisions.

## 3.2 `TACO` implementation

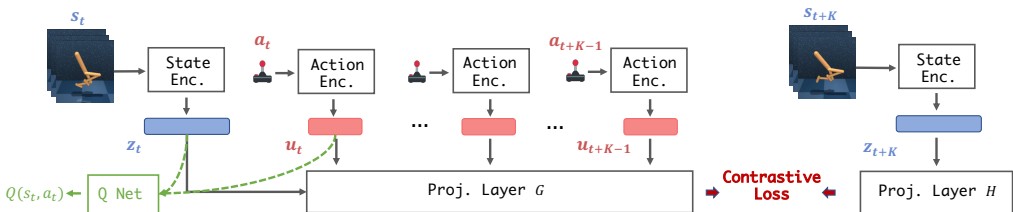

**Figure 2:** A demonstration of our temporal contrastive loss: Given a batch of state-action transition triples $\{(s_t^{(i)}, [a_t^{(i)}, ..., a_{t+K-1}^{(i)}], s_{t+K}^{(i)})\}_{i=1}^N$, we first apply the state encoder and action encoder to get latent state-action encodings: $\{(z_t^{(i)}, [u_t^{(i)}, ..., u_{t+K-1}^{(i)}], z_{t+K}^{(i)})\}_{i=1}^N$. Then we apply two different projection layers to map $(z_t^{(i)}, [u_t^{(i)}, ..., u_{t+K-1}^{(i)}])$ and $z_{t+K}^{(i)}$ into the shared contrastive embedding space. Finally, we learn to predict the correct pairings between $(z_t, [u_t, ..., u_{t+K-1}])$ and $z_{t+K}$ using an InfoNCE loss.

Here we provide a detailed description of the practical implementation of `TACO`. In Figure 2, we illustrate the architecture design of `TACO`. Our approach minimally adapts a base RL algorithm by incorporating the temporal contrastive loss as an auxiliary loss during the batch update process. Specifically, given a batch of state and action sequence transitions $\{(s_t^{(i)}, [a_t^{(i)}, ..., a_{t'-1}^{(i)}], s_{t'}^{(i)})\}_{i=1}^N$ $(t' = t + K)$, we optimize:

$$\mathcal{J}_{\text{TACO}}(\phi, \psi, W, G_\theta, H_\theta) = -\frac{1}{N} \sum_{i=1}^N \log \frac{g_t^{(i)^\top} W h_{t'}^{(i)}}{\sum_{j=1}^N g_t^{(i)^\top} W h_{t'}^{(j)}} \tag{3}$$

Here let $z_t^{(i)} = \phi(s_t^{(i)})$, $u_t^{(i)} = \psi(a_t^{(i)})$ be state and action embeddings respectively. $g_t^{(i)} = G_\theta(z_t^{(i)}, u_t^{(i)}, ..., u_{t'-1}^{(i)})$, and $h_t^{(i)} = H_\theta(z_{t'}^{(i)})$, where $G_\theta$ and $H_\theta$ denote two learnable projection layers that map the latent state $z_t^{(i)}$ as well as latent state and action sequence $(z_t^{(i)}, u_t^{(i)}, ..., u_{t'-1}^{(i)})$ to a common contrastive embedding space. $W$ is a learnable parameter providing a similarity measure between $g_i$ and $h_j$ in the shared contrastive embedding space. Subsequently, both state and action representations are fed into the agent's $Q$ network, allowing the agent to effectively reason about the long-term effects of their actions and better leverage their past experience through state-action abstractions.

In addition to the main `TACO` objective, in our practical implementation, we find that the inclusion of two auxiliary objectives yields further enhancement in the algorithm's overall performance. The first is the CURL [33] loss:

$$\mathcal{J}_{\text{CURL}}(\phi, \psi, W, H_\theta) = -\frac{1}{N} \sum_{i=1}^N \log \frac{h_t^{(i)^\top} W h_t^{(i)^+}}{h_t^{(i)^\top} W h_t^{(i)^+} + \sum_{j \neq i} h_t^{(i)^\top} W h_t^{(j)}} \tag{4}$$

Here, $h_t^{(i)^+} = H_\theta(\phi(s_i^{(t)^+}))$, where $s_i^{(t)^+}$ is the augmented view of $s_i^{(t)}$ by applying the same random shift augmentation as DrQ-v2 [52]. $W$ and $H_\theta$ share the same weight as the ones in `TACO` objective. The third objective is reward prediction:

$$\mathcal{J}_{\text{Reward}}(\phi, \psi, \hat{R}_\theta) = \sum_{i=1}^N \left(\hat{R}_\theta(z_t^{(i)}, u_t^{(i)}, ..., u_{t'-1}^{(i)}) - r^{(i)}\right)^2 \tag{5}$$

Here $r^{(i)} = \sum_{j=t}^{t'-1} r_j^{(i)}$ is the sum of reward from timestep $t$ to $t'-1$, and $\hat{R}_\theta$ is a reward prediction layer. For our final objective, we combine the three losses together with equal weight. As verified in Section 4.1, although `TACO` serves as the central objective that drives notable performance improvements, the inclusion of both CURL and reward prediction loss can further improve the algorithm's performance.

We have opted to use DrQ-v2 [52] for the backbone algorithm of `TACO`, although in principle, `TACO` could be incorporated into any visual RL algorithms. `TACO` extends DrQ-v2 with minimal additional hyperparameter tuning. The only additional hyperparameter is the selection of the prediction horizon $K$. Throughout our experiments, we have limited our choice of $K$ to either 1 or 3, depending on the nature of the environment. We refer the readers to Appendix A for a discussion on the choice of $K$.

## 4 Experiments and results

This section provides an overview of our empirical evaluation, conducted in both online and offline RL settings. To evaluate our approach under online RL, we apply `TACO` to a set of nine challenging visual continuous control tasks from Deepmind Control Suite (DMC) [46]. Meanwhile, for offline RL, we combine `TACO` with existing offline RL methods and test the performance on four DMC tasks, using three pre-collected datasets that differ in the quality of their data collection policies.

### 4.1 Comparison between `TACO` and strong baselines in online RL tasks

**Environment Settings**: In our online RL experiment, we first evaluate the performance of `TACO` on nine challenging visual continuous control tasks from Deepmind Control Suite [46]: Quadruped Run, Quadruped Walk, Hopper Hop, Reacher Hard, Walker Run, Acrobot Swingup, Cheetah Run, Finger Turn Hard, and Reach Duplo. These tasks demand the agent to acquire and exhibit complex motor skills and present challenges such as delayed and sparse reward. As a result, these tasks have not been fully mastered by previous visual RL algorithms, and require the agent to learn an effective policy that balances exploration and exploitation while coping with the challenges presented by the tasks.

In addition to Deepmind Control Suite, we also present the results of `TACO` on additional six challenging robotic manipulation tasks from Meta-world [57]: Hammer, Assembly, Disassemble, Stick Pull, Pick Place Wall, Hand Insert. Unlike the DeepMind Control Suite, which primarily concentrates on locomotion tasks, Meta-world domain provides tasks that involve complex manipulation and interaction tasks. This sets it apart by representing a different set of challenges, emphasizing precision and control in fine motor tasks rather than broader locomotion skills. In Appendix G, we provide a visualization for each Meta-world task.

**Baselines**: We compare `TACO` with four model-free visual RL algorithms **CURL** [33], **DrQ** [53], **DrQ-v2** [52], and **A-LIX** [5]. **A-LIX** builds on **DrQ-v2** by adding adaptive regularization to the encoder's gradients. While `TACO` could also extend **A-LIX**, our reproduction of results from its open-source implementation does not consistently surpass **DrQ-v2**. As such, we do not choose **A-LIX** as the backbone algorithm for `TACO`. Additionally, we compare with two state-of-the-art model-based RL algorithms for visual continuous control, **Dreamer-v3** [18] and **TDMPC** [21], which learn world models in latent space and select actions using either model-predictive control or a learned policy.

`TACO` **achieves a significantly better sample efficiency and performance compared with SOTA visual RL algorithm.** The efficacy of `TACO` is evident from the findings presented in Figure 4 (DMC) , Table 1 (DMC), and Figure 5 (Meta-world). In contrast to preceding model-free visual RL algorithms, `TACO` exhibits considerably improved sample efficiency. For example, on the challenging Reacher Hard task, `TACO` achieves optimal performance in just 0.75 million environment steps, whereas DrQ-v2 requires approximately 1.5 million steps. When trained with only 1 million environment steps, `TACO` on average achieves 40% better performance, and it is even better than the model-based visual RL algorithms Dreamer-v3 and TDMPC on 6 out of 9 tasks. In addition, on more demanding tasks such as Quadruped Run, Hopper Hop, Walker Run, and Cheetah Run, `TACO` continues to outshine competitors, exhibiting superior overall performance after two or three million steps, as illustrated in Figure 4. For robotic manipulation tasks, as shown in Figure 5, `TACO` also significantly outperform the baseline model-free visual RL algorithms, highlighting the broad applicability of `TACO`.

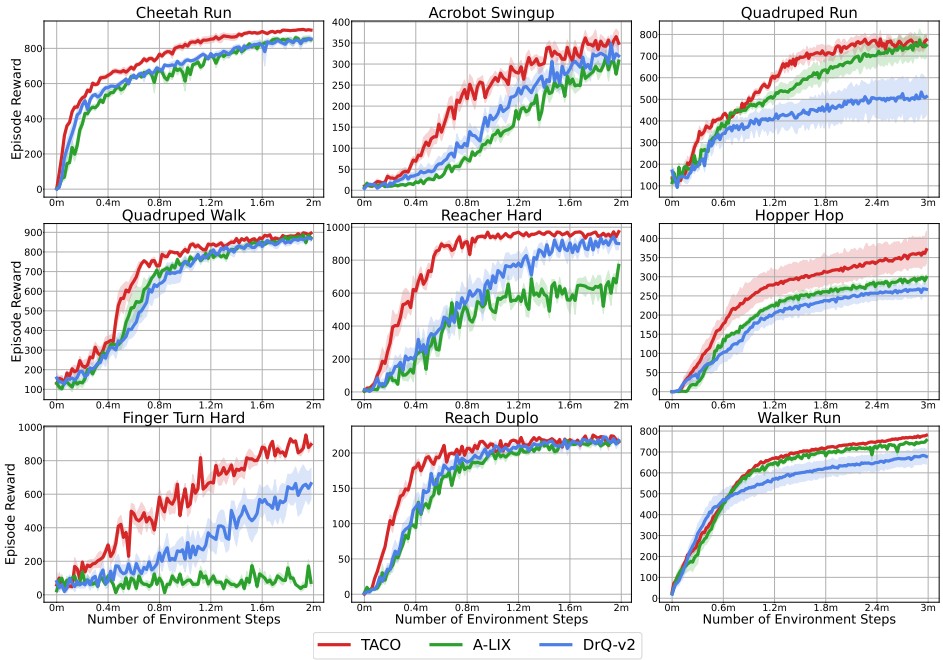

**Figure 4:** (**Deepmind Control Suite**) Performance of `TACO` against two strongest model-free visual RL baselines. Results of DrQ-v2 and A-LIX are reproduced from their open-source implementations, and all results are averaged over 6 random seeds.

**Table 1:** Episode reward of `TACO` and SOTA visual RL algorithms on the image-based DMControl 1M benchmark. Results are averaged over 6 random seeds. Within the table, entries shaded represent the best performance of model-free algorithms, while text **in bold** signifies the highest performance across all baseline algorithms, including model-based algorithms.

| | *Model-free* | | | | | *Model-based* | |
| --- | --- | --- | --- | --- | --- | --- | --- |
| Environment (1M Steps) | TACO | **DrQv2** | **A-LIX** | **DrQ** | **CURL** | **Dreamer-v3** | **TDMPC** |
| Quadruped Run | $\mathbf{541 \pm 38}$ | $407 \pm 21$ | $454 \pm 42$ | $179 \pm 18$ | $181 \pm 14$ | $331 \pm 42$ | $397 \pm 37$ |
| Hopper Hop | $261 \pm 52$ | $189 \pm 35$ | $225 \pm 13$ | $192 \pm 41$ | $152 \pm 34$ | $\mathbf{369 \pm 21}$ | $195 \pm 18$ |
| Walker Run | $637 \pm 11$ | $517 \pm 43$ | $617 \pm 12$ | $451 \pm 73$ | $387 \pm 24$ | $\mathbf{765 \pm 32}$ | $600 \pm 28$ |
| Quadruped Walk | $\mathbf{793 \pm 8}$ | $680 \pm 52$ | $560 \pm 175$ | $120 \pm 17$ | $123 \pm 11$ | $353 \pm 27$ | $435 \pm 16$ |
| Cheetah Run | $\mathbf{821 \pm 48}$ | $691 \pm 42$ | $676 \pm 41$ | $474 \pm 32$ | $657 \pm 35$ | $728 \pm 32$ | $565 \pm 61$ |
| Finger Turn Hard | $632 \pm 75$ | $220 \pm 21$ | $62 \pm 54$ | $91 \pm 9$ | $215 \pm 17$ | $\mathbf{810 \pm 58}$ | $400 \pm 113$ |
| Acrobot Swingup | $\mathbf{241 \pm 21}$ | $128 \pm 8$ | $112 \pm 23$ | $24 \pm 8$ | $5 \pm 1$ | $210 \pm 12$ | $224 \pm 20$ |
| Reacher Hard | $\mathbf{883 \pm 63}$ | $572 \pm 51$ | $510 \pm 16$ | $471 \pm 45$ | $400 \pm 29$ | $499 \pm 51$ | $485 \pm 31$ |
| Reach Duplo | $\mathbf{234 \pm 21}$ | $206 \pm 32$ | $199 \pm 14$ | $36 \pm 7$ | $8 \pm 1$ | $119 \pm 30$ | $117 \pm 12$ |

**Concurrently learning state and action representation is crucial for the success of `TACO`.** To demonstrate the effectiveness of action representation learning in `TACO`, we evaluate its performance on a subset of 4 difficult benchmark tasks and compare it with a baseline method without action representation, as shown in Figure 3. The empirical results underscore the efficacy of the temporal contrastive learning objectives, even in the absence of action representation. For instance, `TACO` records an enhancement of 18% on Quadruped Run and a substantial 51% on Reacher Hard, while the remaining tasks showcase a performance comparable to DrQ-v2. Furthermore, when comparing against `TACO` without action representation, `TACO` achieves a

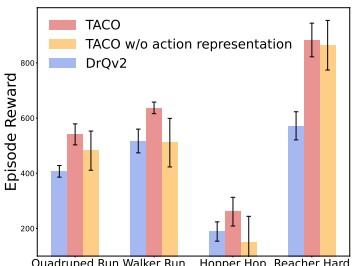

**Figure 3:** 1M Performance of `TACO` with and without action representation

consistent performance gain, ranging from 12.2% on Quadruped Run to a significant 70.4% on Hopper Hop. These results not only emphasize the inherent value of the temporal contrastive learning objective in `TACO`, but also underscore the instrumental role of high-quality action representation in bolstering the performance of the underlying RL algorithms.

`TACO` **learns action representations that group semantically similar actions together.** To verify that indeed our learned action representation has grouped semantically similar actions together, we conduct an experiment within the Cheetah Run task. We artificially add 20 dimensions to the action

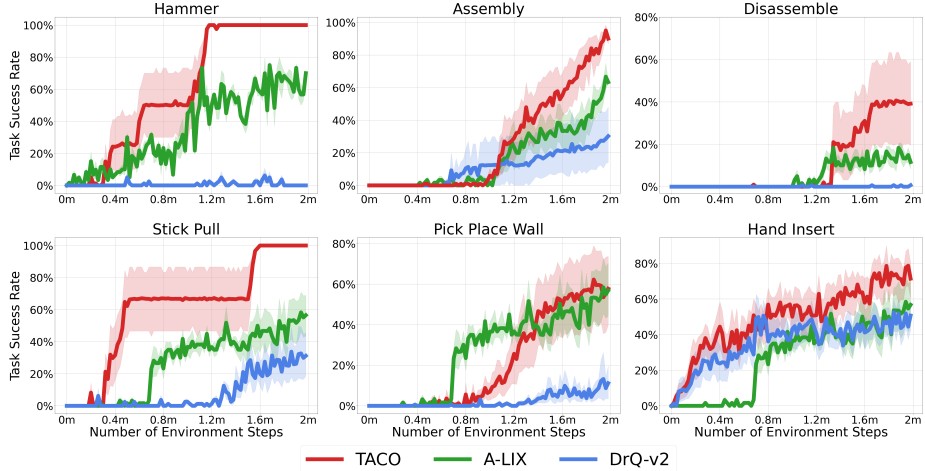

**Figure 5:** (**Meta-world**) Performance of TACO against DrQ-v2 and A-LIX. All results are averaged over 6 random seeds.

space of task Cheetah Run, although only the first six were utilized in environmental interactions. We first train an agent with TACO online to obtain the action representation. Then we select four actions within the original action space $a_1, a_2, a_3, a_4$ to act as centroids. For each of the four centroids, we generate 1000 augmented actions by adding standard Gaussian noises to the last 20 dimensions. We aim to determine if our action representation could disregard these "noisy" dimensions while retaining the information of the first six. Using t-SNE for visualization, we embed the 4000 actions before and after applying action representation. As shown in Figure 6, indeed our learned action representation could group the four clusters, demonstrating the ability of our action representation to extract control relevant information from the raw action space.

**The effectiveness of our temporal contrastive loss is enhanced with a larger batch size.** As is widely acknowledged in contrastive learning research [8, 22, 40, 13], our contrastive loss sees significant benefits from utilizing a larger batch size. In Figure 7a, we illustrate the performance of our algorithms alongside DrQv2 after one million environment steps on the Quadruped Run task. As evident from the plot, batch size greatly influences the performance of our algorithm, while DrQ-v2's baseline performance remains fairly consistent throughout

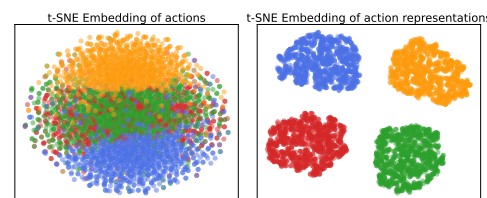

**Figure 6: Left**: t-SNE embedding of actions with distracting dimensions. **Right**: t-SNE embedding of latent representations for actions with distracting dimensions.

training. In order to strike a balance between time efficiency and performance, we opt for a batch size of 1024, which is 4 times larger than the 256 batch size employed in DrQ-v2, but 4 times smaller than the 4096 which is commonly used in the contrastive learning literature [8, 22, 13]. For an analysis on how batch size affects the algorithm's runtime, we direct the reader to Appendix B.

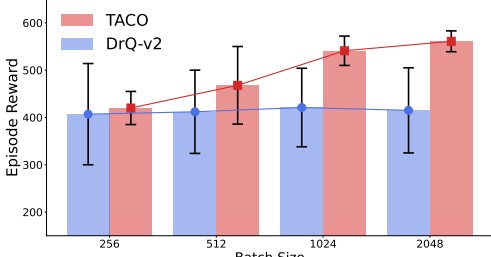

**(a)** TACO and DrQ-v2 across different batch sizes.

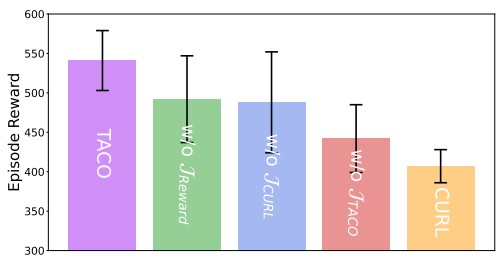

**(b)** TACO with different learning objective removed

**Figure 7:** TACO with different batch sizes and with different component of its learning objectives removed

**Reward prediction and CURL loss serves an auxiliary role to further improve the performance of TACO, while the temporal contrastive loss of TACO is the most crucial component.** In the practical deployment of TACO, two additional objectives, namely reward prediction and CURL loss,

are incorporated to enhance the algorithm's performance. In Figure 7b, we remove one objective at a time on the Quadruped Run task to assess its individual impact on the performance after one million environment steps. As illustrated in the figure, the omission of `TACO`' temporal contrastive objective results in the most significant performance drop, emphasizing its critical role in the algorithm's operation. Meanwhile, the auxiliary reward prediction and CURL objectives, although secondary, contribute to performance improvement to some degree.

**InfoNCE-based temporal action-driven contrastive objective in `TACO` outperforms other representation learning objectives including SPR [43], ATC [44], and DRIML [37].** In Table 2, we have showcased a comparison between our approach and other visual RL representation learning objectives such as **SPR**, **ATC**, and **DRIML**. Given that **SPR** and **DRIML** were not initially designed for continuous control tasks, we have re-implemented their learning objectives using the identical backbone algorithm, DrQ-v2. A similar approach was taken for ATC, with their learning objectives also being reimplemented on DrQ-v2 to ensure a fair comparison. (Without the DrQ-v2 backbone algorithm, the performance reproduced by their original implementation is significantly worse.) Furthermore, recognizing the significance of learning action encoding, as discussed earlier, we have integrated action representation learning into all these baselines. Therefore, the model architecture remains consistent across different representation learning objectives, with the sole difference being the design of the temporal contrastive loss. For **DRIML**, given that only the first action of the action sequence is considered in the temporal contrastive loss, `TACO` and **DRIML** differ when the number of steps $K$ is greater than one. Thus, we indicate **N/A** for tasks where we choose $K = 1$ for `TACO`.

**Table 2:** Comparison with other objectives including SPR [42], ATC [44], and DRIML [37]

| Environment | `TACO` | SPR | ATC | DRIML | DrQ-v2 |
|---|---|---|---|---|---|
| Quadruped Run | $\mathbf{541 \pm 38}$ | $448 \pm 79$ | $432 \pm 54$ | **N/A** | $407 \pm 21$ |
| Walker Run | $\mathbf{637 \pm 21}$ | $560 \pm 71$ | $502 \pm 171$ | **N/A** | $517 \pm 43$ |
| Hopper Hop | $\mathbf{261 \pm 52}$ | $154 \pm 10$ | $112 \pm 98$ | $216 \pm 13$ | $192 \pm 41$ |
| Reacher Hard | $\mathbf{883 \pm 63}$ | $711 \pm 92$ | $863 \pm 12$ | $835 \pm 72$ | $572 \pm 51$ |
| Acrobot Swingup | $\mathbf{241 \pm 21}$ | $198 \pm 21$ | $206 \pm 61$ | $222 \pm 39$ | $210 \pm 12$ |

Table 2 showcases that while previous representation learning objectives have proven benefit in assisting the agent to surpass the DrQ-v2 baseline by learning a superior representation, our approach exhibits consistent superiority over other representation learning objectives in all five evaluated environments. These results reinforce our claim that `TACO` is a more effective method for learning state-action representations, allowing agents to reason more efficiently about the long-term outcomes of their actions in the environment.

## 4.2 Combining `TACO` with offline RL algorithms

In this part, we discuss the experimental results of `TACO` within the context of offline reinforcement learning, emphasizing the benefits our temporal contrastive state/action representation learning objective brings to visual offline RL. Offline visual reinforcement learning poses unique challenges, as algorithms must learn an optimal policy solely from a fixed dataset without further interaction with the environment. This necessitates that the agent effectively generalizes from limited data while handling high-dimensional visual inputs. The state/action representation learning objective of `TACO` plays a vital role in addressing these challenges by capturing essential information about the environment's dynamics, thereby enabling more efficient generalization and improved performance. `TACO` can be easily integrated as a plug-and-play module on top of existing strong offline RL methods, such as **TD3+BC** [11] and **CQL** [31].

For evaluation, we select four challenging visual control tasks from DMC: Hopper Hop, Cheetah Run, Walker Run, and Quadruped Run. For each task, we generate three types of datasets. The **medium** dataset consists of trajectories collected by a single policy of medium performance. The precise definition of "medium performance" is task-dependent but generally represents an intermediate level of mastery, which is neither too poor nor too proficient. The **medium-replay** dataset contains trajectories randomly sampled from the online learning agent's replay buffer before it reaches a medium performance level. The **full-replay** dataset includes trajectories randomly sampled throughout the online learning phase, from the beginning until convergence. The dataset size for Walker, Hopper, and Cheetah is 100K, while for the more challenging Quadruped Run task, a larger dataset size of 500K is used to account for the increased difficulty. We compute the normalized reward by diving the offline RL reward by the best reward we get during online `TACO` training.

**Table 3:** Offline Performance (Normalized Reward) for different offline RL methods. Results are averaged over 6 random seeds. $\pm$ captures the standard deviation over seeds.

| Task/ Dataset | | TD3+BC w. TACO | TD3+BC | CQL w. TACO | CQL | DT | IQL | BC |
|---|---|---|---|---|---|---|---|---|
| Hopper Hop | Medium | **52.4 ± 0.4** | 51.2 ± 0.8 | **47.9 ± 0.6** | 46.7 ± 0.2 | 40.5 ± 3.6 | 2.0 ± 1.7 | 48.2 ± 0.8 |
| | Medium-replay | **67.2 ± 0.1** | 62.9 ± 0.1 | **74.6 ± 0.4** | 68.7 ± 0.1 | 65.3 ± 1.8 | 57.6 ± 1.4 | 25.9 ± 3.2 |
| | Full-replay | **97.6 ± 1.4** | 83.8 ± 2.3 | **101.2 ± 1.9** | 94.2 ± 2.0 | 92.4 ± 0.3 | 47.7 ± 2.2 | 65.7 ± 2.7 |
| Cheetah Run | Medium | **66.6 ± 0.5** | 66.1 ± 0.3 | **70.1 ± 0.4** | 66.7 ± 1.7 | 64.3 ± 0.7 | 1.7 ± 1.1 | 62.9 ± 0.1 |
| | Medium-replay | **62.6 ± 0.2** | 61.1 ± 0.1 | **72.3 ± 1.2** | 67.3 ± 1.1 | 67.0 ± 0.6 | 26.5 ± 3.2 | 48.0 ± 3.1 |
| | Full-replay | **92.5 ± 2.4** | 91.2 ± 0.8 | **86.9 ± 2.4** | 65.0 ± 3.9 | 89.6 ± 1.4 | 14.6 ± 3.7 | 69.0 ± 0.3 |
| Walker Run | Medium | **49.2 ± 0.5** | 48.0 ± 0.2 | **49.6 ± 1.0** | 49.4 ± 0.9 | 47.3 ± 0.3 | 4.4 ± 0.4 | 46.2 ± 0.6 |
| | Medium-replay | **63.1 ± 0.6** | 62.3 ± 0.2 | **62.3 ± 2.6** | 59.9 ± 0.9 | 61.7 ± 1.1 | 41.4 ± 2.8 | 18.5 ± 0.8 |
| | Full-replay | **86.8 ± 0.6** | 84.0 ± 1.6 | **88.1 ± 0.1** | 79.8 ± 0.6 | 81.6 ± 0.8 | 18.1 ± 3.7 | 30.8 ± 1.8 |
| Quadruped Run | Medium | **60.6 ± 0.1** | 60.0 ± 0.2 | **58.1 ± 3.7** | 55.9 ± 9.1 | 14.6 ± 3.8 | 0.8 ± 0.8 | 56.2 ± 1.1 |
| | Medium-replay | **61.3 ± 0.3** | 58.1 ± 0.5 | **61.9 ± 0.2** | 61.2 ± 0.9 | 19.5 ± 2.2 | 58.4 ± 4.4 | 51.6 ± 3.3 |
| | Full-replay | **92.6 ± 0.7** | 89.3 ± 0.4 | **92.1 ± 0.1** | 85.2 ± 2.5 | 14.5 ± 1.1 | 36.3 ± 5.9 | 57.6 ± 0.7 |
| **Average Normalized Score** | | **71.0** | 68.2 | **72.1** | 66.7 | 48.4 | 25.8 | 54.9 |

We compare the performance of **TD3+BC** and **CQL** with and without TACO on our benchmark. Additionally, we also compare with the decision transformer (**DT**) [7], a strong model-free offline RL baseline that casts the problem of RL as conditional sequence modeling, **IQL** [30], another commonly used offline RL algorithm, and the behavior cloning (**BC**) baseline. For **TD3+BC**, **CQL** and **IQL**, which were originally proposed to solve offline RL with vector inputs, we add the their learning objective on top of DrQ-v2 to handle image inputs.

Table 3 provides the normalized reward for each dataset. The results underscore that when combined with the strongest baselines **TD3+BC** and **CQL**, TACO achieves consistent performance improvements across all tasks and datasets, setting new state-of-the-art results for offline visual reinforcement learning. This is true for both the medium dataset collected with a single policy and narrow data distribution, as well as the medium-replay and replay datasets with a diverse distribution.

## 5 Related work

### 5.1 Contrastive learning in visual reinforcement learning

Contrastive learning has emerged as a powerful technique for learning effective representations across various domains, particularly in computer vision [47, 8, 22, 23, 49]. This success is attributed to its ability to learn meaningful embeddings by contrasting similar and dissimilar data samples. In visual reinforcement learning, it's used as a self-supervised auxiliary task to improve state representation learning, with InfoNCE [47] being a popular learning objective. In CURL [33], it treats augmented states as positive pairs, but it neglects the temporal dependency of MDP. CPC [47], ST-DIM [2], and ATC [44] integrate temporal relationships into the contrastive loss by maximizing mutual information between current state representations (or state histories encoded by LSTM in CPC) and future state representations. However, they do not consider actions, making positive relationships in the learning objective policy-dependent. DRIML [37] addresses this by maximizing mutual information between state-action pairs at the current time step and the resulting future state, but its objective remains policy-dependent as it only provides the first action of the action sequence. Besides, ADAT [29] and ACO [59] incorporate actions into contrastive loss by labeling observations with similar policy action outputs as positive samples, but these methods do not naturally extend to tasks with non-trivial continuous action spaces. A common downside of these approaches is the potential for unstable encoder updates due to policy-dependent positive relations. In contrast, TACO is theoretically sufficient, and it tackles the additional challenge of continuous control tasks by simultaneously learning state and action representations.

In addition to the InfoNCE objective, other self-supervised learning objective is also proposed. Approahces such as DeepMDP [12], SPR [42], SGI [43], and EfficientZero [56] direct learn a latent-space transition model. Notably, these methods predominantly target Atari games characterized by their small, well-represented, and abstract discrete action spaces. When dealing with continuous control tasks, which often involve a continuous and potentially high-dimensional action space, the relationships between actions and states become increasingly intricate. This complexity poses a significant challenge in effectively capturing the underlying dynamics. In contrast, by framing the latent dynamics model predictions as a self-supervised InfoNCE objective, the mutual information

guided approach used by `TACO` is better suited for continuous control task, resulting in more stable optimization and thus better state and action representations.

## 5.2 Action representation in reinforcement learning

Although state or observation representations are the main focus of prior research, there also exists discussion on the benefits and effects of learning action representations. Chandak et al. [6] propose to learn a policy over latent action space and transform the latent actions into actual actions, which enables generalization over large action sets. Allshire et al. [1] introduce a variational encoder-decoder model to learn disentangled action representation, improving the sample efficiency of policy learning. In model-based RL, strategies to achieve more precise and stable model-based planning or roll-out are essential. To this end, Park et al. [39] propose an approach to train an environment model in the learned latent action space. In addition, action representation also has the potential to improve multi-task learning [25], where latent actions can be shared and enhance generalization.

## 6 Conclusion

In this paper, we have introduced a conceptually simple temporal action-driven contrastive learning objective that simultaneously learns the state and action representations for image-based continuous control — `TACO`. Theoretically sound, `TACO` has demonstrated significant practical superiority by outperforming SOTA online visual RL algorithms. Additionally, it can be seamlessly integrated as a plug-in module to enhance the performance of existing offline RL algorithms. Despite the promising results, `TACO` does present limitations, particularly its need for large batch sizes due to the inherent nature of the contrastive InfoNCE objective, which impacts computational efficiency. Moving forward, we envisage two primary directions for future research. Firstly, the creation of more advanced temporal contrastive InfoNCE objectives that can function effectively with smaller data batches may mitigate the concerns related to computational efficiency. Secondly, the implementation of a distributed version of `TACO`, akin to the strategies employed for DDPG in previous works [3, 24], could significantly enhance training speed. These approaches offer promising avenues for further advancements in visual RL.

## 7 Acknowledgement

Zheng, Wang, Sun and Huang are supported by National Science Foundation NSF-IIS-FAI program, DOD-ONR-Office of Naval Research, DOD Air Force Office of Scientific Research, DOD-DARPA-Defense Advanced Research Projects Agency Guaranteeing AI Robustness against Deception (GARD), Adobe, Capital One and JP Morgan faculty fellowships.

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

# Appendix

## A   Effects of the choice of prediction horizon $K$

`TACO` functions as a versatile add-on to current online and offline RL algorithms, requiring only the prediction horizon $K$ as an additional hyperparameter. In our investigations, we select $K$ as either 1 or 3 across all tasks. The performance of `TACO` with $K$ values of 1 and 3 is compared against the DrQ-v2 baselines in Figure 8. As illustrated in the figure, `TACO` outperforms the baseline DrQ-v2 for both $K = 1$ and 3, indicating the effectiveness of the temporal contrastive loss in concurrent state and action representation learning. In comparing $K = 1$ to $K = 3$, we observe that a longer prediction horizon ($K = 3$) yields superior results for four out of nine tasks, specifically Hopper Hop, Quadruped Walk, Acrobot Swingup, and Reacher Hard. Conversely, for the Quadruped Run and Walker Run tasks, a shorter temporal contrastive loss interval ($K = 1$) proves more beneficial. For the remaining tasks, the choice between $K = 1$ and $K = 3$ appears to have no discernable impact.

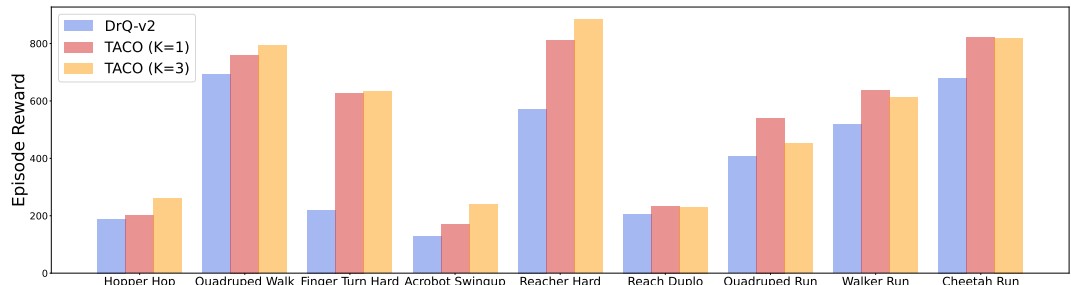

**Figure 8:** 1M Performance of `TACO` with step size $K = 1$ and $K = 3$.

Our speculation hinges on the rate of changes in the agent's observations. While Theorem 3.1 applies regardless of the prediction horizon $K$, a shorter horizon (such as $K = 1$) can offer a somewhat shortsighted perspective. This becomes less beneficial in continuous control tasks where state transitions within brief time intervals are negligible. Therefore, in environments exhibiting abrupt state transitions, a larger $K$ would be advantageous, whereas a smaller $K$ would suffice for environments with gradual state transitions.

To substantiate this conjecture, we conduct a simple experiment on the task Hopper Hop where we aim to simulate the different rates of changes in the agent's observations. We verify this by varying the size of action repeats, which correspond to how many times a chosen action repeats per environmental step. Consequently, a larger action repeat size induces more pronounced observational changes. In our prior experiments, following the settings of DrQ-v2, we fix the size of the action repeat to be 2. In this experiment, we alter the size of the action repeat for Hopper Hop to be 1, 2, and 4. For each action repeat size, we then compare the 1M performance of `TACO` with the prediction horizon $K$ selected from the set $\{1, 3, 5\}$. Interestingly, as demonstrated in Figure 9, the optimal $K$ values for different action repeats are reversed: 5 for an action repeat of 1, 3 for an action repeat of 2, and 1 for an action repeat of 4. This observation further substantiates our assertion that the optimal choice of prediction horizon correlates with the rate of change in environmental dynamics.

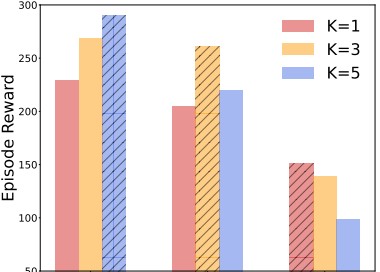

**Figure 9:** 1M Performance of `TACO` with different prediction horizon $K$ under different action repeat sizes. Shaded columns correspond to the best prediction horizon $K$ under a fixed action repeat size.

## B   Time-efficiency of `TACO`

In this section, we compare the time efficiency of different visual RL algorithms. In Table 4, we present a comprehensive comparison of the speed of these algorithms, measured in frames per second

(**FPS**). Additionally, we also provide **FPS** for `TACO` with different batch sizes in Table 5 as a reference. To ensure a fair comparison, all the algorithms are tested on Nvidia A100 GPUs.

As mentioned in §4.1, the InfoNCE objective in `TACO` requires a large batch size of 1024, which is 4 times larger than the batch size used in DrQ-v2. Consequently, this increases the processing time, making our method to run about 3.6 times slower than DrQ-v2, with similar time efficiency as DrQ. Therefore, the primary limitation of our method is this time inefficiency caused by the usage of large batch sizes. Potential solutions to improve the time efficiency could involve the implementation of a distributed version of our method to speed up training. We intend to explore these improvements in future work related to our paper.

**Table 4:** Frames per second (**FPS**) for visual RL algorithms. **B** stands for batch size used in their original implementations.

| TACO (**B**:1024) | DrQv2 (**B**:256) | A-LIX (**B**:256) | DrQ (**B**:512) | CURL (**B**:512) | Dreamer-v3 (**B**:256) | TDMPC (**B**:256) |
|---|---|---|---|---|---|---|
| 35 | 130 | 98 | 33 | 20 | 31 | 22 |

**Table 5:** Frames per second (**FPS**) for `TACO` with different batch size

| TACO (**B**:1024) | TACO (**B**:512) | TACO (**B**:256) |
|---|---|---|
| 35 | 65 | 94 |

# C   Experiment details

## C.1   Online RL

In this section, we describe the implementation details of `TACO` for the online RL experiments. We implement `TACO` on top of the released open-source implementation of DrQ-v2, where we interleave the update of `TACO` objective with the original actor and critic update of DrQ-v2. Below we demonstrate the pseudo-code of the new update function.

```
### Extract feature representation for state and actions.
def update(batch):
    obs, action_sequence, reward, next_obs = batch
    ### Update Agent's critic function
    update_critic(obs, action_sequence, reward, next_obs)
    ### Update the agent's value function
    update_actor(obs)
    ### Update TACO loss
    update_taco(obs, action_sequence, reward, next_obs)
```

**Listing 1:** Pytorch-like pseudo-code how `TACO` is incorporated into the update function of existing visual RL algorithms.

Next, we demonstrate the pseudo-code of how `TACO` objective is computed with pytorch-like pseudocode.

```
# state_encoder:     State/Observation Encoder (CNN)
# action_encoder:    Action Encoder (MLP with 1-hidden layer)
# sequence_encoder: Action Sequence encoder (Linear Layer)
# reward_predictor: Reward Prediction Layer (MLP with 1-hidden layer)
# G:                 Projection Layer I (MLP with 1-hidden layer)
# H:                 Projection Layer II (MLP with 1-hidden layer)
# aug:               Data Augmentation Function (Radnom Shift)
# W:                 Matrix for computing similarity score

def compute_taco_objective(obs, action_sequence, reward, next_obs):
    ### Compute feature representation for both state and actions.
    z       = state_encoder(aug(obs))
    z_anchor = state_encoder(aug(obs), stop_grad=True)
    next_z = state_encoder(aug(next_obs), stop_grad=True)
    u_seq  = sequence_encoder(action_encoder(action_sequence))
    ### Project to joint contrastive embedding space
    x  = G(torch.cat([z, u_seq], dim=-1))
    y  = H(next_z)
    ### Compute bilinear product x^TWy
```

```
20    ### Diagonal entries of x^TWy correspond to positive pairs
21    logits = torch.matmul(x, torch.matmul(W, y.T))
22    logits = logits - torch.max(logits, 1)
23    labels = torch.arange(n)
24    taco_loss = cross_entropy_loss(logits, labels)
25
26    ### Compute CURL loss
27    x = H(z)
28    y = H(z_anchor).detach()
29    logits = torch.matmul(x, torch.matmul(W, y.T))
30    logits = logits - torch.max(logits, 1)
31    labels = torch.arange(n)
32    curl_loss = cross_entropy_loss(logits, labels)
33
34    ### Reward Prediction Loss
35    reward_pred = reward_predictor(z, u_seq)
36    reward_loss = torch.mse_loss(reward_pred, reward)
37
38    return taco_loss + curl_loss + reward_loss
```

**Listing 2:** Pytorch-like pseudo-code for how `TACO` objective is computed

Then when we compute the Q values for both actor and critic updates, we use the trained state and action encoder. Same as what is used in DrQ-v2, we use 1024 for the hidden dimension of all the encoder layers and 50 for the feature dimension of state representation. For action representation, we choose the dimensionality of the action encoding, which corresponds to the output size of the action encoding layer, as $\lceil 1.25 \times |\mathcal{A}| \rceil$. In practice, we find this works well as it effectively extracts relevant control information from the raw action space while minimizing the inclusion of irrelevant control information in the representation. See Appendix D for an additional experiment on testing the robustness of `TACO` against the dimensionality of action representation.

## C.2 Offline RL

TD3+BC, CQL, IQL all are originally proposed for vector-input. We modify these algorithms on top of DrQ-v2 so that they are able to deal with image observations. For TD3+BC, the behavior cloning regularizer is incorporated into the actor function update, with the regularizer weight $\alpha_{\text{TD3+BC}} = 2.5$ as defined and used in Scott et al. [11]. In our experiments, no significant performance difference was found for $\alpha$ within the $[2,3]$ range. In the case of CQL, we augment the original critic loss with a Q-value regularizer and choose the Q-regularizer weight, $\alpha_{\text{CQL}}$, from $\{0.5, 1, 2, 4, 8\}$. Table 6 presents the chosen $\alpha_{\text{CQL}}$ for each dataset.

**Table 6:** Hyperparameter of $\alpha_{\text{CQL}}$ used in different tasks/datasets.

| Task/ Dataset | | $\alpha_{\text{CQL}}$ |
|---|---|---|
| Hopper Hop | Medium | 0.5 |
| | Medium-replay | 0.5 |
| | Replay | 2 |
| Cheetah Run | Medium | 0.5 |
| | Medium-replay | 2 |
| | Replay | 4 |
| Walker Run | Medium | 0.5 |
| | Medium-replay | 1 |
| | Replay | 4 |
| Quadruped Run | Medium | 0.5 |
| | Medium-replay | 2 |
| | Replay | 4 |

For IQL, we adopt the update functions of both policy and value function from its open-source JAX implementation into DrQ-v2, setting the inverse temperature $\beta$ to 0.1, and $\tau = 0.7$ for the expectile. Lastly, for the Decision Transformer (DT), we adapt from the original open-source implementation by Chen et al. [7] and use a context length of 20.

## D  Sensitivity of `TACO` to action representation dimensionality: an additional experiment

In all of our previous experiments, we use $\lceil 1.25 \times |\mathcal{A}| \rceil$ as the latent dimensions of the action space for `TACO`. In practice, we find this works well so that it retains the rich information of the raw actions while being able to group semantically similar actions together. To test the algorithm's sensitivity to this hyperparameter, we conduct an experiment on the Quadruped Run task, which has a 12-dimensional action space. In Figure 10, we show the 1M performance of `TACO` with different choices of latent action dimensions. Notably, we observe that as long as the dimensionality of the action space is neither too small (6-dimensional), which could limit the ability of latent actions to capture sufficient information from the raw actions, nor too large (24-dimensional), which might introduce excessive control-irrelevant information, the performance of `TACO` remains robust to the choice of action representation dimensionality.

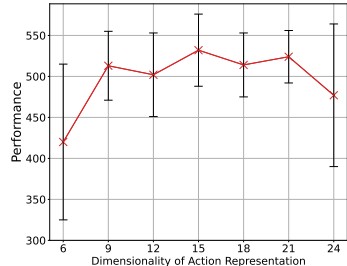

**Figure 10:** 1M Performance of `TACO` with different dimensionality of latent action representations on Quadruped Run ($|\mathcal{A}| = 12$). Error bar represents standard deviation across 8 random seeds.

## E  Sensitivity of `TACO` vs. DrQ-v2 to random seeds

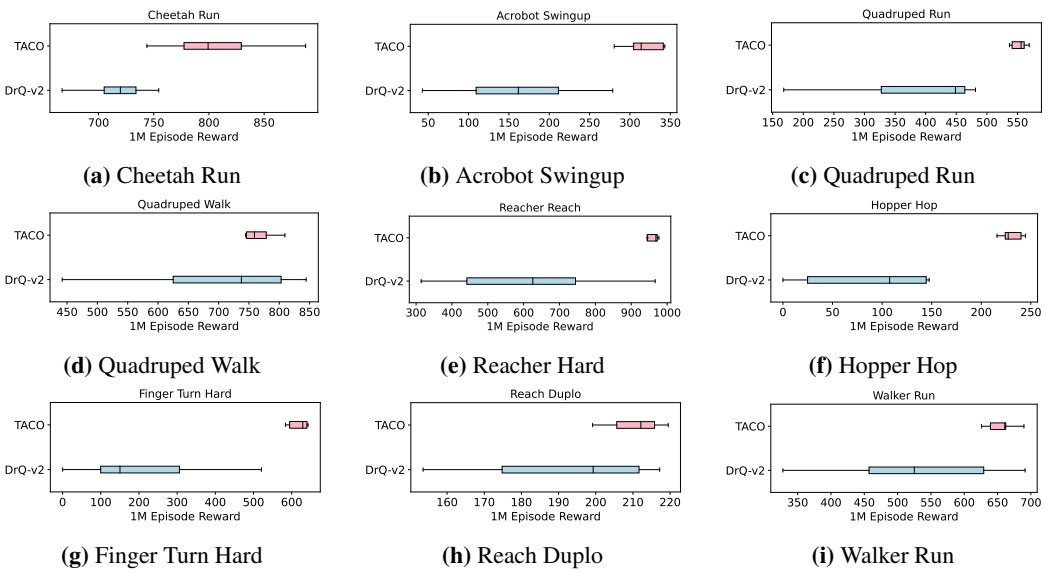

**(a)** Cheetah Run    **(b)** Acrobot Swingup    **(c)** Quadruped Run

**(d)** Quadruped Walk    **(e)** Reacher Hard    **(f)** Hopper Hop

**(g)** Finger Turn Hard    **(h)** Reach Duplo    **(i)** Walker Run

**Figure 11:** Robustness analysis of `TACO` vs. DrQ-v2 across multiple (6) random seeds

In this section, we conduct a comprehensive analysis of the algorithm's robustness under random seed, comparing the performance of our proposed method, `TACO`, with the baseline algorithm, DrQ-v2, across multiple seeds on nine distinct tasks shown in Figure 4. Our aim is to gain insights into the stability and robustness of both algorithms under varying random seeds, providing valuable information to assess their reliability.

Remarkably, Figure 11 demonstrates the impressive robustness of `TACO` compared to DrQ-v2. `TACO` consistently demonstrates smaller performance variations across different seeds, outperforming the baseline on almost every task. In contrast, runs of DrQ-v2 frequently encounter broken seeds. These results provide strong evidence that `TACO` not only improves average performance but also significantly enhances the robustness of the training process, making it more resilient against failure cases. This enhanced robustness is crucial for real-world applications, where stability and consistency are essential for successful deployment.

# F Tackling the complex "Manipulator Bring Ball" task: an additional online RL experiment within the DeepMind Control Suite

In addition to the nine tasks depicted in Figure Figure 4, we conduct an additional experiment on an even more complex task "Manipulator Bring Ball". This task necessitates the robotic arm to precisely locate, grab, and transport the ball to a specified goal location. A visualization of a successful trajectory is illustrated in Figure 12a.

The challenge of this domain lies in the sparse reward structure, the demand for accurate arm control, and the need for temporally extended skills. As a result, despite training for 30 million frames, as demonstrated in Figure 12b, DrQ-v2 fails to successfully complete this task. In comparison, TACO manages to successfully solve this task for half of the random initializations, i.e., 3 out of 6 seeds.

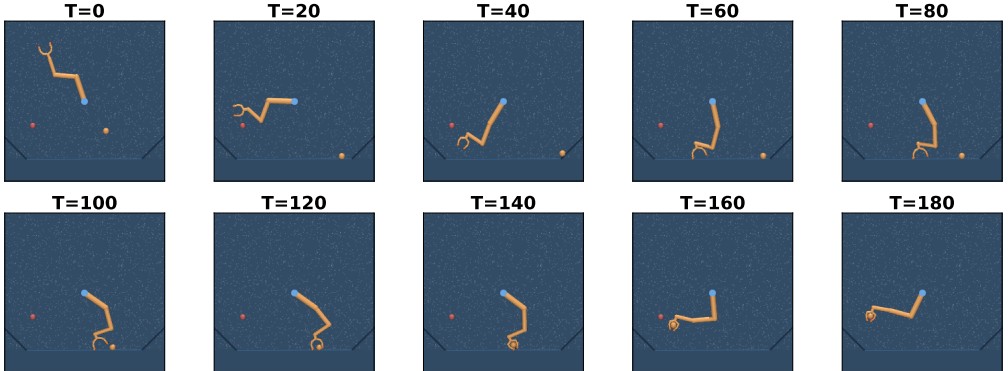

**(a)** Visualization of an expert trajectory on manipulator bring ball.

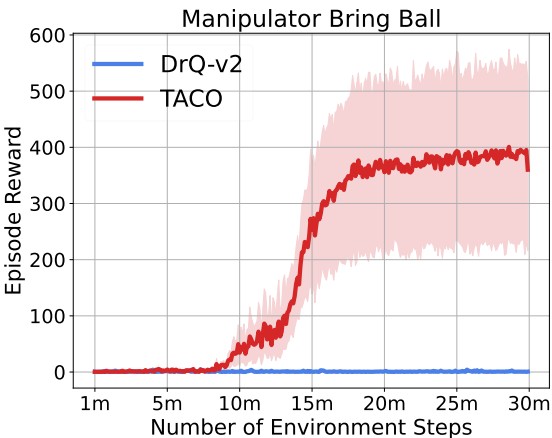

**(b)** Online learning curves on manipulator bring ball aggregated over 6 random seeds.

**Figure 12:** Manipulator Bring Ball Task

# G   Visualization of Meta-world Tasks

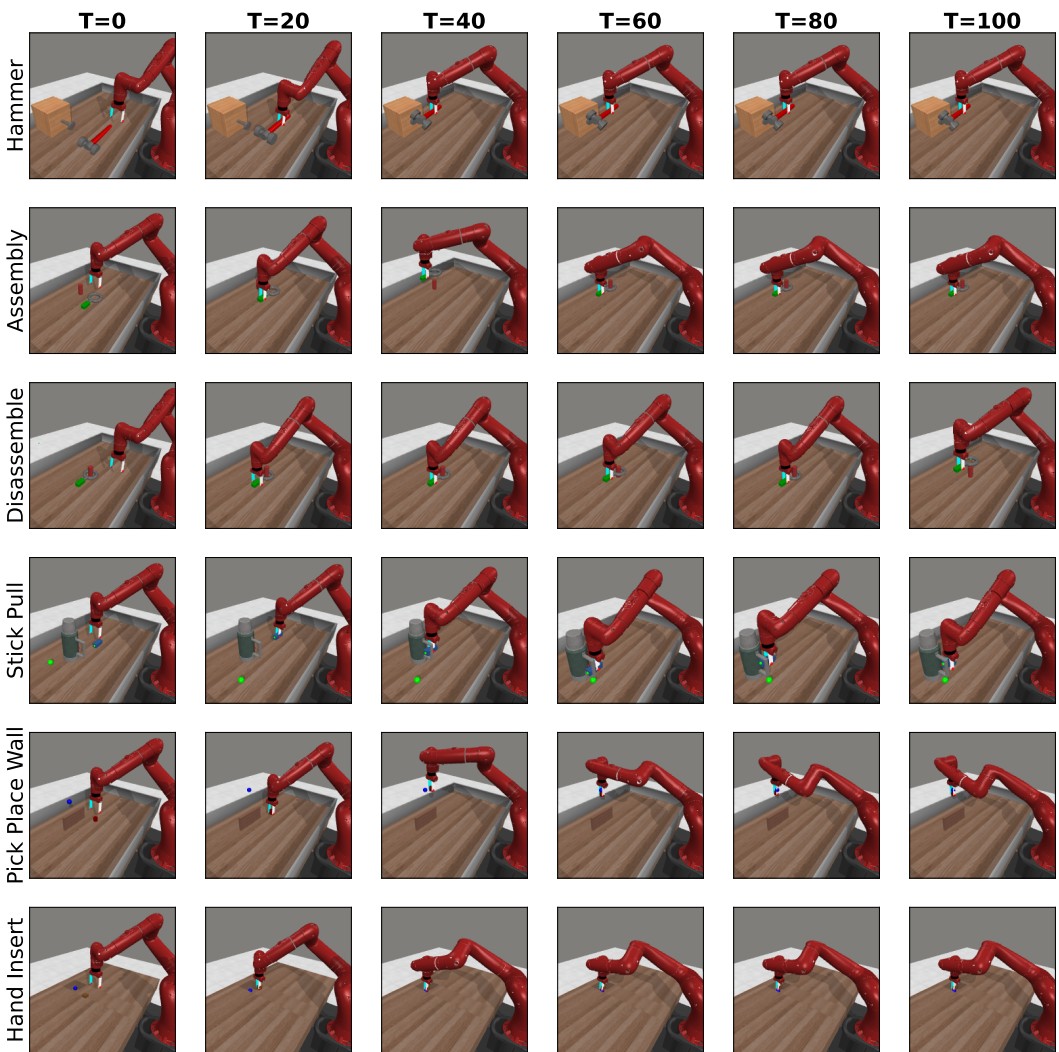

**Figure 13:** Visualization of expert trajectories for each Meta-world task

# H   Proof of Theorem 3.1

In this section, we prove Theorem 3.1, extending the results of Rakely et al. [41]. We will use notation $A_{t:t+K}$ to denote the action sequence $A_t, ..., A_{t+K}$ from timestep $t$ to $t+K$, and we will use $U_{t:t+K}$ to denote the sequence of latent actions $U_t, ..., U_{t+K}$ from timestep $t$ to $t+K$.

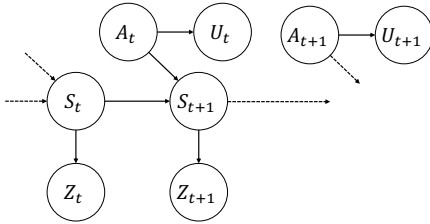

**Figure 14:** The graphical diagram

**Proposition H.1.** *Let $X$ be the return to go* $\displaystyle\sum_{i=0}^{H-t-K} \gamma^i R_{t+K+i}$*, with the conditional independence assumptions implied by the graphical model in Figure 14. If $I(Z_{t+K}; Z_t, U_{t:t+K-1}) = I(S_{t+K}; S_t, A_{t:t+K-1})$, then $I(X; Z_t, U_{t:t+K-1}) = I(X; S_t, A_{t:t+K-1})$*

*Proof.* We prove by contradictions. Suppose there exists a pair of state action representation $\phi_Z, \psi_U$ and a reward function $r$ such that $I(Z_{t+K}; Z_t, U_{t:t+K-1}) = I(S_{t+K}; S_t, A_{t:t+K-1})$, but $I(X; Z_t, U_{t:t+K-1}) < I(X; S_t, U_{t:t+K-1}) < I(X; S_t, A_{t:t+K-1})$. Then it suffices to show that $I(S_{t+k}; Z_t, U_{t:t+K-1}) < I(S_{t+K}; S_t, U_{t:t+K-1})$, which would give us the desired contradiction since $I(Z_{t+k}; Z_t, U_{t:t+K-1}) \leq I(S_{t+k}; Z_t, U_{t:t+K-1})$, and $I(S_{t+K}; S_t, U_{t:t+K-1}) \leq I(S_{t+K}; S_t, A_{t:t+K-1})$.

Now we look at $I(X; Z_t, S_t, U_{t:t+K-1})$. Apply chain rule of mutual information:

$$I(X; Z_t, S_t, U_{t:t+K-1}) = I(= X; Z_t|S_t, U_{t:t+K-1}) + I(X; S_t, U_{t:t+K-1}) \tag{6}$$
$$= 0 + I(X; S_t, U_{t:t+K-1}) \tag{7}$$

Applying the chain rule in another way, we get

$$I(X; Z_t, S_t, U_{t:t+K-1}) = I(X; S_t|Z_t, U_{t:t+K-1}) + I(X; Z_t, U_{t:t+K-1}) \tag{8}$$

Therefore, we get

$$I(X; S_t, U_{t:t+K-1}) = I(X; S_t|Z_t, U_{t:t+K-1}) + I(X; Z_t, U_{t:t+K-1}) \tag{9}$$

By our assumption that $I(X; Z_t, U_{t:t+K-1}) < I(X; S_t, U_{t:t+K-1})$, we must have

$$I(X; S_t|Z_t, U_{t:t+K-1}) > 0 \tag{10}$$

Next, we expand $I(S_{t+K}; Z_t, S_t, U_{t:t+K-1})$:

$$I(S_{t+K}; Z_t, S_t, U_{t:t+K-1}) = I(S_{t+K}; Z_t|S_t, U_{t:t+K-1}) + I(S_{t+K}; S_t, U_{t:t+K-1}) \tag{11}$$
$$= 0 + I(S_{t+K}; S_t, U_{t:t+K-1}) \tag{12}$$

On the other hand, we have

$$I(S_{t+K}; Z_t, S_t, U_{t:t+K-1}) = I(S_{t+K}; S_t|Z_t, U_{t:t+K-1}) + I(S_{t+K}; Z_t, U_{t:t+K-1}) \tag{13}$$
$$\tag{14}$$

Thus we have

$$I(S_{t+K}; S_t|Z_t, U_{t:t+K-1}) + I(S_{t+K}; Z_t, U_{t:t+K-1}) = I(S_{t+K}; S_t, U_{t:t+K-1}) \tag{15}$$

But then because $I(S_{t+K}; S_t|Z_t, U_{t:t+K-1}) > I(X; S_t|Z_t, U_{t:t+K-1})$ as $S_t \to S_{t+K} \to X$ forms a Markov chain, it is greater than zero by the Inequality (10). As a result, $I(S_{t+K}; Z_t, U_{t:t+K-1}) < I(S_{t+K}; S_t, U_{t:t+K-1}) < I(S_{t+K}; S_t, A_{t:t+K-1})$. This is exactly the contradiction that we would like to show.

Before proving Theorem 3.1, we need to cite another proposition, which is proved as Lemma 2 in Rakely et al. [41]

**Proposition H.2.** *Let $X, Y.Z$ be random variables. Suppose $I(Y,Z) = I(Y,X)$ and $Y \perp Z|X$, then $\exists p(Z|X)$ s.t. $\forall x, p(Y|X = x) = \int p(Y|Z)p(Z|X = x)dz$*

*Proof of Theorem 3.1*: Based on the graphical model, it is clear that

$$\max_{\phi,\psi} I(Z_{t+K}, [Z_t, U_t, ..., U_{t+K-1}]) = I(S_{t+K}; [S_t, A_t, ..., A_{t+K-1}]) \tag{16}$$

Now define the random variable of return-to-go $\overline{R}_t$ such that

$$\overline{R}_t = \sum_{k=0}^{H-t} \gamma^k R_{t+k} \tag{17}$$

Based on Proposition H.1, because

$$I(Z_{t+K}; Z_t, U_{t:t+K-1}) = I(S_{t+K}; S_t, A_{t:t+K-1})$$

we could conclude that

$$I(\overline{R}_{t+K}; Z_t, U_{t:t+K-1}) = I(\overline{R}_{t+K}; S_t, A_{t:t+K-1}) \tag{18}$$

Now applying Proposition H.2, we get

$$\mathbb{E}_{p(z_t, u_{t:t+K-1}|S_t=s, A_{t:t+K-1}=a_{t:t+K-1})}[p(\overline{R}_t|Z_t, U_{t:t+K-1})] = p(\overline{R}_t|S_t = s, A_{t:t+K-1}) \tag{19}$$

As a result, when $K = 1$, for any reward function $r$, given a state-action pair $(s_1, a_1)$, $(s_2, a_2)$ such that $\phi(s_1) = \phi(s_2), \psi(a_1) = \psi(a_2)$, we have $Q_r(s_1, a_1) = \mathbb{E}_{p(\overline{R}_t|S_t=s_1, A_t=a_1)}[\overline{R}_t] = \mathbb{E}_{p(\overline{R}_t|S_t=s_2, A_t=a_2)}[\overline{R}_t]$. This is because $p(\overline{R}_t|S_t = s_1, A_t = a_1) = p(\overline{R}_t|S_t = s_2, A_t = a_2)$ by Equation (18) as $p(z_t|S_t = s_1) = p(z_t|S_t = s_2), p(u_t|A_t = a_1) = p(u_t|A_t = a_2)$. In case when $K > 1$, because if $\mathbb{E}[Z_{t+K}, [Z_t, U_t, ..., U_{t+K-1}]] = \mathbb{E}[S_{t+K}, [S_t, A_t, ..., A_{t+K-1}]]$, then for any $1 \leq k \leq K$, $\mathbb{E}[Z_{t+k}, [Z_t, U_t, ..., U_{t+k-1}]] = \mathbb{E}[S_{t+k}, [S_t, A_t, ..., A_{t+k-1}]]$, including $K = 1$, by Data processing Inequality. (Intuitively, this implies that if the information about the transition dynamics at a specific step is lost, the mutual information decreases as the timestep progresses, making it impossible to reach its maximum value at horizon $K$.) Then the same argument should also apply here.

# I  Additional related work

## I.1  Visual reinforcement learning

In this paper, we focus primarily on visual-control tasks, and this section reviews relevant prior work in visual RL. For visual-control environments, representation learning has shown to be a key. Many prior works show that learning auxiliary tasks can encourage the representation to be better aligned with the task and thus enhance the performance, such as reconstructing pixel observations [55], minimizing bisimulation distances [58], fitting extra value functions [4, 9], learning latent dynamics models [12, 42, 45], multi-step inverse dynamics model [32, 26] or various control-relevant objectives [27]. Model-based methods are also shown to be successful in efficient visual RL, which learn the transition dynamics based on the encoded observation [16, 35, 15, 56]. Data augmentation can also be used to smooth out the learned representation or value functions to improve the learning performance [54, 20, 19].

## I.2  Additional works on self-supervised/contrastive learning in reinforcement learning

In §5.1, we summarize the works that apply self-supervised/contrastive learning objectives to improve the sample efficiency of visual reinforcement learning. In this section, we discuss the additional works that apply self-supervised/contrastive learning objectives to a broader set of topics in reinforcement learning.

Several recent works have investigated the use of self-supervised/contrastive learning objectives to pre-train representations for reinforcement learning (RL) agents. Parisi et al. [38] propose PVR,

which leverages a pre-trained visual representation from MoCo [22] as the perception module for downstream policy learning. Xiao et al. [50] introduce VIP, a method that pre-trains visual representations using masked autoencoders. Additionally, Ma et al. [36] propose VIP, which formulates the representation learning problem as offline goal-conditioned RL and derives a self-supervised dual goal-conditioned value-function objective.

Besides pretraining state representations, in goal-conditioned RL, the work by Eysenbach et al. [10] establishes a connection between learning representations with a contrastive loss and learning a value function. Moreover, self-supervised learning has also been employed for effective exploration in RL. Guo et al. [14] introduce BYOL-Explore, a method that leverages the self-supervised BYOL objective [13] to acquire a latent forward dynamics model and state representation. The disagreement in the forward dynamics model is then utilized as an intrinsic reward for exploration. Another approach, ProtoRL by Yarats et al. [51], presents a self-supervised framework for learning a state representation through a clustering-based self-supervised learning objective in the reward-free exploration setting.

