# OpenReview forum: "$\texttt{TACO}$: Temporal Latent Action-Driven Contrastive Loss for Visual Reinforcement Learning"
_NeurIPS.cc/2023/Conference — NeurIPS 2023 poster_

### Official Review · Reviewer_oD4S · 2023-06-19

**Soundness:** 3 good
**Presentation:** 3 good
**Contribution:** 2 fair
**Rating:** 5
**Confidence:** 3

**Summary:**

This work proposes a simple yet effective temporal contrastive learning approach for encoding the high-dimensional observations and inputs for reinforcement learning. The authors propose a loss function (TACO) related to the mutual information between representations of current states paired with action sequences and the future states. By jointly optimizing this TACO loss with CURL loss and reward prediction loss, the proposed method outperforms the SOTA representation learning algorithms in on-policy and off-policy RL frameworks. They also validate their effectiveness in model-based frameworks.

**Strengths:**

- The proposed method outperforms the SOTA algorithms both in model-free and model-based frameworks
- The experiments were conducted in various environments with multiple baselines (including on-policy, off-policy, and model-based).
- An appropriate ablation study was performed for each loss term.

**Weaknesses:**

- In terms of TACO loss, the difference between DRIML and the proposed method lies in the fact that the proposed method considers the entire action sequence up to t+K, rather than just observing single action. Looking at Figure 7 in the supplementary materials, it can be seen that in 4 out of 9 environments, TACO performs better when K=1 compared to K=3. This could suggest that extending DRIML appropriately to a continuous action space yields performance similar to TACO.
- The authors mention that they implemented some baselines based on DrQ-v2 and additionally considered action as an input (lines 258-260). However, there is no detailed explanation of this implementation, making it somewhat difficult to ensure a fair comparison.

**Questions:**

- Regarding “recognizing the significance of learning action encoding, as discussed earlier, we’ve integrated action representation learning into all these baselines.” could you provide a more detailed explanation of how action inputs were considered in ATC and SPR in the existing methodologies?
- In lines 257-258, it is stated, "Without the DrQ-v2 backbone algorithm, the performance reproduced by their original implementation is significantly worse." What is the difference between extending DRIML to a continuous action space and DRIML with the DrQ-v2 backbone you used as a baseline? Which of these differences do you think has the most significant impact on performance?
- Does the RL loss flow into the encoder, or does the encoder only update through the loss in Equation 3-5? If it is the latter, how does the RL loss affect performance?
- Does the performance decrease if the action encoder is not used and only projection is performed?
- Line 318: Please check the name of the algorithm (ACO → DBC for [50]).

**Limitations:**

The authors mentioned the limitation in the conclusion section. Since they utilize InfoNCE objective and the performance of the proposed method depends on the batch size, the main objective of TACO impacts computational efficiency.

---

> ### Author Rebuttal · Authors · 2023-08-08
>
> Thank you for your insightful feedback and review! Below we address the concerns and questions that you have raised. We are encouraged that you appreciate TACO's significant empirical performance and recognize the comprehensiveness of our experiments in both online and offline RL, alongside the ablation studies for each loss term.
>
> ---
>
> We would like to first explain the question around the implementation of other representation learning baselines.
> > The authors mention that they implemented some baselines based on DrQ-v2 and additionally considered action as an input. However, there is no detailed explanation of this implementation, making it somewhat difficult to ensure a fair comparison.
> > Could you provide a more detailed explanation of how action inputs were considered in ATC and SPR in the existing methodologies?
>
> The learning objective of ATC loss does not involve action input. For SPR, it is implemented as follows.
>
> ```
> z_hat = state_encoding(observation[0])
> spr_loss = 0
> for k in range(K):
>     u = action_encoding(action[k])
>     z_hat = h(torch.concat([z_hat,u], dim=-1)) ### transition model
>     z_next = state_encoding_target(observation[k+1]) ### target encoder
>     y_hat = q(g(z_hat))
>     y_next = g_target(z_hat)
>     spr_loss += -cosine_similarity(y_hat, y_next)
> ```
> Here, h is a latent transition model, g is a projection layer, and q is the prediction head. Following the insights of TACO, we let the critic and SPR loss share the same acton encoder so that we only focus on the comparison of the temporal contrastive loss. In other words, we update action encoder from both SPR loss as well as the critic's TD loss. (See my comments below for a detailed discussion on this design choice for TACO.)
>
> ---
>
> > TACO performs better when K=1 compared to K=3. This could suggest that extending DRIML appropriately to a continuous action space yields performance similar to TACO.
> > What is the difference between extending DRIML to a continuous action space and DRIML with the DrQ-v2 backbone you used as a baseline?
>
> We would like to clarify the key distinction between our approach and DRIML, as well as other representation learning baselines.
>
> 1. The original DRIML and SPR papers, build their methods on top of the C51 algorithm (DQN for SPR) and is specifically tailored to environments with discrete action spaces. C51 algorithm itself cannot extend to continuous action spaces. This is why we choose to re-implement DRIML on top of DrQ-v2, a simple yet strong online RL algorithm for visual continuous control.
> 2. DRIML focuses on environments with small, well-represented, abstract discrete action spaces, overlooking the importance of action representation learning.
> 3. In contrast, we identify the importance of action representaion learning in continuous control, an under-explored topic in previous works. We introduce TACO as a simple yet effecive approach to utilize temporal contrastive loss to learn state and action representations.
> 4. For the comparison with DRIML in Table 2, we also incorporate action representation learning in the same way as TACO and only focus on the comparison of the design of temporal contrastive loss.
> 5. Non-trivial improvement on Hopepr Hop and Acrobot Swingup: We still observe notable enhancements in tasks such as hopper hop (showing a 20.8% improvement) and acrobot swingup (with an 8.5% increase). This can be attributed to DRIML's contrastive loss positive pairs being policy-dependent, potentially causing stability concerns during policy updates. Contrarily, TACO's design is policy-neutral, offering a more stable solution.
>
> In summary, the main contribution of our paper lies in both identifying the importance of action representation in continuous control and proposing TACO as a simple yet effective solution to address this problem.
>
>
> ---
>
> > Does the RL loss flow into the encoder, or does the encoder only update through the loss in Equation 3-5? If it is the latter, how does the RL loss affect performance?
>
> The action encoder is updated through both the critic's TD loss and the TACO loss, allowing it to learn more informative action representations. Specifically, when we exclude the TD loss from updating the action encoder, there is a noticeable performance drop, with the 1M online performance falling from 541 +- 38 to 492 +- 44 on Quadruped Run, and from 261 +- 52 to 211 +- 86 on Hopper Hop.
>
> ---
>
> > Does the performance decrease if the action encoder is not used and only projection is performed?
>
> Yes, the performance does indeed decrease if the action encoder is not used and only projection is performed. By excluding the action encoder and taking raw action as input for the critic, the 1M online performance drops to 499 +- 32 on Quadruped Run and to 221 +- 36 on Hopper Hop, down from 541 +- 38 and 261 +- 52, respectively. Compared to DrQ-v2's 1M performance of 407 +- 21 for Quadruped Run and 192 +- 41 for Hopper Hop, we still observe a noticeable improvement as state and action representations benefit from the temporal contrastive loss. Nevertheless, these findings emphasize the importance of the action encoder and suggest that allowing the critic and temporal contrastive loss to share the same action embedding effectively aids in learning action representations. We appreciate the reviewer's insight and question, and we will clarify this point in our revised manuscript.

---

> > ### Comment · Reviewer_oD4S · 2023-08-16
> > **Response to authors**
> >
> > I appreciate the authors for their detailed responses to my questions. However, there are some parts of the authors' response that are a bit difficult to understand. I'll write down the parts I understood, and please let me know if the following explanations are incorrect.
> >
> > 1. The only difference between DRIML with the DrQ-v2 and TACO is whether k is 1 or can take non-1 values.
> > 2. Additionally, among the 5 environments shown in Table 2, TACO performs best when K=1 in 2 of them. (Looking at Figure 7 in the supplementary materials, it can be seen that in 4 out of 9 environments, TACO performs best when K=1.)
> > 3. In other words, simply changing from C51 to DrQ-v2 in DRIML already achieves SOTA-level performance in 4 out of the 9 environments.
> >
> > If the above content is correct, considering that the contribution of DRIML or TACO is not about which RL agent to use but rather in the context of representation learning, it might suggest a significant limitation in the novelty of TACO.

---

> > > ### Author Response · Authors · 2023-08-16
> > > **Response**
> > >
> > > Thank you for your response. We believe there may be some misunderstandings regarding our previous response. In Table 2 of our original manuscript, when comparing TACO to DRIML, we did not just simply adopt the DRIML objective from C51 to DrQ-v2 for continuous actions. We also incorporated insights from TACO, allowing DRIML to learn an action representation jointly from both the temporal contrastive loss and the critic's TD loss. As shown in our manuscript and earlier response, this important insight has led to a significant performance boost for TACO. We also incorporated it into the implementation of the baseline DRIML algorithm because our focus here was solely on comparing the design of the temporal contrastive loss. Yet, as illustrated in Table 2, we observed non-trivial improvements in tasks like Hopper Hop (a 20.8% increase) and Acrobot Swing-up (an 8.5% rise). As explained in our earlier response, this is potentially due to the inherent limitations in the design of the temporal contrastive loss in DRIML, which results in the positive relationships being policy-dependent and unstable.

---

> > > > ### Comment · Reviewer_oD4S · 2023-08-19
> > > >
> > > > I appreciate the authors for their detailed responses to my questions. I keep my score.

---

### Official Review · Reviewer_Y2ee · 2023-07-05

**Soundness:** 3 good
**Presentation:** 3 good
**Contribution:** 2 fair
**Rating:** 7
**Confidence:** 4

**Summary:**

This paper introduces an auxiliary objective based on contrastive learning to learn action and state representation for continuous control benchmarks. The auxiliary objective is called TACO. The main idea behind the objective is to maximize the mutual information between the current state s_t, current and future actions {a_t, a_{t + 1}, … , a_{t + k}} and the future state s_{t + k}. They show that they can outperform various model-free, model-based methods on various environment in the deepmind control suite.


**Strengths:**

The main thing I like about the paper is the breadth of applicability of the approach. The authors use the objective for both model-free and offline RL methods. The method is compared against relevant baselines in both these cases. The authors also compare against other relevant methods that use auxiliary pretaining objectives for learning state representations. This thorough comparison shows the effectiveness of the approach.


**Weaknesses:**

The main concern I have is the use of reward prediction and curl objectives. The main novelty of the paper seems to be the contrastive objective in equation 3. But it seems that there are no experiments that evaluates and studies this objective in isolation. Figure 6b removes one objective at a time but it would be nice to study the effective of removing both reward prediction and curl objectives. Without this it is hard to say whether the empirical gains are justified by the motivation and theoretical claims of the paper. I would be happy to increase the score if this point is addressed.


**Questions:**

It seems that the TACO objective is used in conjunction with the underlying RL algorithm and not as a pertaining step. I wonder if the authors have tried using it for pretraining? If not, is there any reason why the authors have not used it for pretraining?

I could not find the architectural details of H_\theta and G_\theta, could the authors specify the details of these?


**Limitations:**

The authors have discussed limitations.

---

> ### Author Rebuttal · Authors · 2023-08-08
>
> Thank you for your detailed feedback and review! We are encouraged by your recognition of the broad applicability of our approach, as manifested in our application of TACO for both model-free and offline visual RL settings.
>
> ---
> To address your question of CURL and reward prediction loss, below we conduct a comprehensive experiment by running TACO without CURL and reward prediction loss on five tasks (same as the five tasks in **Table 2** of our original manuscript): Quadruped Run, Hopper Hop, Walker Run, Reacher Hard, Acrobot Swingup. Here we show the 1M online RL performance.
>
> |               | TACO     | TACO w/o reward & CURL| DrQv2|
> | -----------    | ----     | ----------- |----------- |
> | Quadruped Run  | 541 +- 38|  501 +- 24  | 407 +- 21|
> | Walker Run     | 637 +- 21|  615 +- 11  | 517 +- 43|
> | Hopper Hop     | 261 +- 38| 242 +- 12   | 192 +- 41|
> | Reacher Hard   | 883 +- 21| 882 +-  67  | 572 +- 51|
> | Acrobot Swingup| 241 +- 21| 301 +-  42  | 210 +- 12|
>
> As the experimental result suggests, for initial test tasks like Quadruped Run, Walker Run, and Hopper Hop, omitting reward prediction and CURL loss seems to have a slight negative influence on performance, albeit not substantial. This observation led us to incorporate these two auxiliary losses into our final objective. Interestingly, we also found that in the Acrobot Swingup task, the agent's 1M performance actually improved when these two losses were removed, increasing from 241 to 301.
> These experimental findings reinforce our claim that while CURL and reward prediction loss could further improve the performance in many tasks, the proposed temporal contrastive loss of TACO is indeed the central and most impactful component.
>
> ---
>
> Next, we would like to address your two other questions.
>
> > Question1: I wonder if the authors have tried using it for pretraining?
>
> Exploring the use of TACO loss as a self-supervised pretraining objective for both state and action representation is indeed an exciting future direction. In the current paper, our primary focus lies in online and offline RL learning from scratch without any prior knowledge. Recognizing the interest in this aspect, and despite the limited rebuttal period, we conduct an additional experiment to highlight the potential of using TACO for pretraining these representations. The detailed results can be found in **Figure 18** of our attached PDF.
>
> Specifically, we experimented with pretraining on an offline dataset of **Walker Walk Replay** (generated in a manner consistent with our "Replay" dataset in offline RL experiments) to test the learned representation's generalization to a new task, **Walker Run**, evaluating with both online RL and few-shot behavior cloning. For online RL, we initialize DrQ-v2 with the pretained state and action encoders. For few-shot behavior cloning, we initialize the policy with the pretrained state encoder. As demonstrated in **Figure 18** of the attached 1-page PDF, by pretraining on the walker walk dataset, the state and action representation indeed captures the essential information for the shared embodiment (walker) across two different tasks. Thus, it facilitates both efficient online RL training and few-shot imitation learning.
>
> ---
>
> > Question 2: I could not find the architectural details of H_\theta and G_\theta, could the authors specify the details of these?
>
> G is a two-layer Multilayer Perceptron (MLP) where the input size is observation feature dimension plus K (number of timesteps in TACO) times the latent action dimension. The output size of G is the same as the observation feature dimension. H is also a two-layer MLP, with both its input and output sizes being the observation feature dimension. Both G and H utilize a hidden layer size of 1024, and same as DrQ-v2, the observation feature dimension used in TACO is 50.

---

> > ### Author Response · Authors · 2023-08-16
> > **Additional Questions?**
> >
> > Thank you again for your constructive feedback. In our earlier response, we provided an additional ablation study on the CURL and reward prediction loss, as well as another experiment using TACO as a feature representation pretraining objective. If you have further questions, we are more than happy to answer them.

---

> > > ### Comment · Reviewer_Y2ee · 2023-08-16
> > >
> > > Thanks for your response. I have updated my score to reflect the rebuttal.

---

### Official Review · Reviewer_1TK9 · 2023-07-06

**Soundness:** 3 good
**Presentation:** 3 good
**Contribution:** 2 fair
**Rating:** 6
**Confidence:** 4

**Summary:**

The paper introduces TACO, a framework that learns state and action representations simultaneously in visual reinforcement learning for continuous control tasks. TACO optimizes mutual information between current state-action pairs and future state representations. It additionally optimizes 2 auxiliary losses. Experimental results demonstrate TACO can achieve great performance gains in online and offline RL settings. TACO offers a flexible and stable approach for capturing essential information in high-dimensional continuous control tasks.

**Strengths:**

- TACO is a simple yet effective framework that learns state and action representations by an auxiliary contrastive learning task. TACO could be integrated into both online and offline visual RL algorithms flexibly.
- Extensive experiments on the DeepMind Control Suite demonstrate that TACO has outstanding performance.
- The paper provides theoretical analysis of TACO's objectives.
- In general, the paper is well-written.

**Weaknesses:**

- The proposed contrastive learning objective is very similar to DRIML. Both methods improve the performance of the model-free agent by enhancing the predictability of the latent representation through contrastive learning. The major difference is that in this paper, the whole action sequence is given instead of the first action only. Therefore, I think the novelty is limited.
- Extra parameter K is required to tune on different tasks.
- It would be better if the proposed method can be evaluated in more environments besides DMC.

**Questions:**

- Line 134-136, "maximizing this mutual information objective ensures that the learned representations are sufficient for making optimal decisions." However, Thereom 3.1 only shows that the optimal Q function is the same with equivalent state and action representations. How do you ensure the optimal action will be the same?
- In Eq. 4, Why do you use a learnable parameter $W$ as a similarity measure, as the features have already been projected to the latent space by network G and H？
- Is the action encoder only used in the temporal contrastive loss or is it also incorporated in policy learning?

**Limitations:**

The computational limitation is considered in the paper. In addition to the time complexity, I am also curious about the GPU memory cost caused by the large batch size.

---

> ### Author Rebuttal · Authors · 2023-08-08
>
> Thank you for your insightful feedback and review! We are encouraged that you recognize TACO's simplicity, flexibility, outstanding performance, and theoretical analysis, all of which contribute to the strength of our approach. Below we address the concerns and questions that you have raised.
>
> ---
>
> > Novelty of our approach: comparison with DRIML
>
> We would like to clarify the key distinction between our approach and DRIML, as well as other representation learning baselines. Below we list the comparison with DRIML in bullet points.
>
> 1. DRIML & SPR builds on top of C51/DQN algorithm which do not extend to environments with continuous action space.
> 2. DRIML & SPR focus on environments with small, well-represented, abstract discrete action spaces, overlooking the importance of action representation learning.
> 3. In contrast, we identify the importance of action representaion learning in continuous control, an under-explored topic in previous works. We introduce TACO as a simple yet effecive approach to utilize temporal contrastive loss to learn state and action representations.
> 4. For the comparison with DRIML in Table 2, we also incorporate action representation learning in the same way as TACO and only focus on the comparison of the design of temporal contrastive loss.
> 5. Non-trivial improvement on Hopepr Hop and Acrobot Swingup: We still observe notable enhancements in tasks such as hopper hop (showing a 20.8% improvement) and acrobot swingup (with an 8.5% increase). This can be attributed to DRIML's contrastive loss positive pairs being policy-dependent, potentially causing stability concerns during policy updates. Contrarily, TACO's design is policy-neutral, offering a more stable solution.
>
> In summary, the main contribution of our paper lies in both identifying the importance of action representation in continuous control and proposing TACO as a simple yet effective solution to address this problem.
>
> ---
>
> > Extra parameter K is required to tune
>
> We ackowledge that this is one limitation of TACO. However, we would like to point out that in TACO, we only use K=1 or K=3 in all of our experiments. Thus, it requires minimum hyperparameter tuning efforts to find the best K.
>
>
> ---
>
> > It would be better if the proposed method can be evaluated in more environments besides DMC.
>
> We appreciate your suggestion to evaluate our method in diverse environments. In response, we have selected six challenging tasks within the **Meta-world** domain to test the online learning performance of TACO against the DrQ-v2 baseline. The learning curve, presented in **Figure 17** of the attached PDF file, once again demonstrates TACO's significant performance improvement. It successfully solves complex tasks such as hammer, assembly, disassemble, stick pull, and pick place wall, which DrQ-v2 cannot accomplish within 2 million online interaction steps. These results underscore that the insights of TACO extend beyond the locomotion tasks of the DeepMind Control Suite to complex and intricate robotic manipulation tasks.
>
> ---
>
> > How do you ensure the optimal action will be the same?
> >
> In lines 132-134, we're not claiming that the optimal action will be identical. The result indicates that if two state-action pairs possess the same state and action representations, their optimal Q values coincide. Thus, the optimal value function can be factorized as $Q^*(\phi(s), \psi(a))$. This means for a given state, if two actions share the same representation, they'll have an equal optimal Q value. Thus, optimal actions do not have to be unique.
>
> ---
>
> > In Eq. 4, Why do you use a learnable parameter W as a similarity measure, as the features have already been projected to the latent space by network G and H？
>
> This is a design choice that we have made, following the same design choice as CURL and CPC. We could instead use the cosine similiarity measure, as done in MoCo, SimCLR, and CLIP. In our empirical evaluation on the quadruped run task, we find that using cosine similarity lead to a 3.5% decrease in performance at the 1M mark (541 ± 38 vs. 522 ± 56).
>
> ---
>
> >  Is the action encoder only used in the temporal contrastive loss or is it also incorporated in policy learning?
>
> The action encoder is utilized both in the temporal contrastive loss and the critic learning, as illustrated in Figure 2. It is updated based on both the TACO loss as well as the Temporal Difference (TD) loss from the critic functions.
>
> ---
>
> > GPU memory cost:
>
> TACO's batch size of 1024 is four times that of DrQ-v2 but also only a quarter of the standard batch size in the contrastive learning literature. Using the same CNN architecture as DrQ-v2, TACO's GPU memory requirement for the Quadruped Run task is 4.8 GB, in contrast to DrQ-v2's 2.2 GB. This fits easily under a single RTX 2080 Ti GPU.

---

> > ### Author Response · Authors · 2023-08-16
> > **Additional Questions?**
> >
> > Thank you again for your constructive feedback. In our earlier response, we clarified our distinction from DRIML and conducted additional experiments on six tasks from the Meta-world domain. If you have further questions, we are more than happy to answer them.

---

> > > ### Comment · Area_Chair_pxx6 · 2023-08-19
> > >
> > > Thanks for your rebuttal. Unless the reviewer responds, I will assume most of their concerns have been addressed.

---

> > > ### Comment · Reviewer_1TK9 · 2023-08-21
> > > **Thank you for the rebuttal**
> > >
> > > I appreciate the authors for the detailed explanation. I think the proposed method makes non-trivial improvements on existing methods and demonstrates strong performances. I have updated my score to 6.

---

### Official Review · Reviewer_gh9U · 2023-07-07

**Soundness:** 3 good
**Presentation:** 3 good
**Contribution:** 3 good
**Rating:** 7
**Confidence:** 3

**Summary:**

This work introduces TACO, a novel state-action representation learning technique based on contrastive learning. Empirically, TACO outperforms both model-free and model-based visual RL baselines in both online and offline settings.

**Strengths:**

1. The work studies joint state-action representation learning, which is less studied than state representation learning.
2. The empirical results for the online setting seem promising and highlights the importance of both state *and* action representation learning.

**Weaknesses:**

1. The core novelty of this work is the contrastive temporal objective for learning state-action representations and section Fig. 6b shows that it is most responsible for the observed performance improvement. However, it's unclear how important TACO is in the offline RL experiments; TACO makes use of data augmentation (for the CURL loss) while the remaining baselines don't have access to augmented data; thus TACO has an unfair advantage. I would like to see ablations similar to Fig. 6b for the offline experiments.
2. None of the baselines considered learn a latent action representation. While Fig. 3 shows that TACO's action representation is important, it's unclear if it's any better than other methods that learn an action representation via e.g. a state-conditioned variational autoencoder [1].

I am willing to raise my score if the authors address these concerns.

**Minor comments:**
1. Fig. 6 a and b should have the same vertical scale to make the figures more easily comparable.



**Questions:**

1. Is there a specific reason why the authors use CURL as an additional auxiliary tasks (as opposed to e.g. SODA [2])? If CURL can be swapped with another method, it would be worth mentioning this.
2. Could the authors clarify the experiment in lines 215-231? You add 20 dummy dimensions to the action vector when learning the action representation, though I'm not sure what is meant by "1000 noise dimensions", and I am not sure where "the 4000 actions" come from.
3. Line 120: Is there a reason why the authors chose to let the action representation be independent of the state?
4. Line 115-120: The goal stated here is to compress the state-action representation while retaining information needed to solve the task. Is the action representation being compressed? The sensitivity analysis in Appendix D considers increasing the action dimensionality in Quadruped Run by a factor of 2.

[1] Laser: Learning a latent action space for efficient reinforcement learning. Allshire et al. ICRA 2021.
[2] Generalization in reinforcement learning by soft data augmentation. Hansen & Wang. ICRA 2021.

**Limitations:**

See Weaknesses.

---

> ### Author Rebuttal · Authors · 2023-08-08
>
> Thank you for your detailed feedback and review! We are encouraged that you recognize the novelty of our approach in joint state-action representation learning and appreciate the promising empirical results of TACO. Below we first address your two concerns.
>
> ---
> **Offline RL ablation**:
> We conduct additional experiments in which we remove the CURL and reward prediction loss from TACO, referring to this modified version as "TACO w/o R&C." In **Figure 16** of the attached PDF, we present the normalized rewards for both CQL and TD3+BC with TACO, TACO w/o R&C, and without TACO. These results underscore that the removal of reward prediction and CURL losses does not significantly diminish TACO's performance, emphasizing that they are not the primary drivers of TACO's superior performance.
>
> **Comparison with LASER**:
> First, LASER and TACO differ significantly in their action representation learning. While TACO uses action representation only for critic learning, LASER alternates between policy/critic learning and action representation learning, potentially causing instability in RL learning, as the agent's latent action space continuously evolves throughout the training process.
>
> LASER does not make their code publicly available, and it builds upon the state-input SAC algorithm without providing comprehensive experimental details. Therefore, we re-implement their learning objectives from scratch based on DrQ-v2. On Walker Run, we evaluated the performance of LASER with different dimensionalities of the learned latent action space.
>
> |    | LASER (dim=4)| LASER (dim=6)| LASER (dim=8) | DrQ-v2 | TACO |
> | -----------   | ----------- | -----------|-----------|-----------|-----------|
> | Walker Run  | 251 +- 79   | 492 +- 51  | 511 +- 41| 517 +- 43| 637 +- 11 |
>
> (Note: the original action space of Walker Run is 6 dimensional and TACO's latent action space dimension is 8.)
>
> Our findings suggest that LASER's performance suffers from the iterative updates between RL and action representation learning, limiting its improvement over DrQ-v2, particularly when compressing the action space. However, we must acknowledge that due to the time constraints for the rebuttal phase, our choice of hyperparameters for LASER may not have been optimal, potentially failing to reproduce their best results.
>
> ---
>
> Next, here is the response to your specific questions.
> >Question 1: Is there a specific reason why the authors use CURL as an additional auxiliary tasks (as opposed to e.g. SODA)?
>
> We chose to use CURL, but we could also apply SODA loss in place of CURL loss. We test TACO with SODA loss on the Quadruped Run and Walker Run tasks, As shown in **Figure 15 (Left)** of the attached PDF, the results suggest that for Quadruped Run, CURL as the auxiliary loss outperforms SODA, but yields comparable performance on Walker Run.
>
> >Question 2: Clarification on Line 215-231 (Figure 5)
>
> The primary objective of this experiment is to assess whether TACO's learned action representation can effectively extract control-relevant action information. To further clarify, here is the Gym-like pseudocode for our modified Cheetah Run environment:
>
> ```
> class CheetahRunNew:
>     def __init__(self, env, action_dim=6, distract_dim=20):
>         self.orginal_env = env
>         self.action_space = Box(-1.0, 1.0, shape=(action_dim + distract_dim))
>         self.action_dim = action_dim
>     def step(self, action):
>         return self.orginal_env.step(action[:self.action_dim])
> ```
> In this modified Cheetah Run environment, while the action space dimensionality has been expanded to 26, only the first 6 dimensions are utilised. We then train TACO in this modified environment, setting the dimensionality of learned latent action embedding to be 6.
> To evaluate whether the learned action representation indeed captures the information of the first 6 dimensions, we sample four actions, $a_1, a_2, a_3, a_4 \in R^6$, to act as centroids. For each of the four centroids, we generate 1000 augmented actions by adding standard Gaussian noises to the last 20 dimensions.
> Our hypothesis is that actions with the same centroid should possess similar latent representations, given that they are fundamentally the same action. To validate this, we generated a t-SNE action embedding plot (**Figure 5** of our original manuscript), which revealed successful clustering of semantically similar actions.
>
> >Question 3: Is there a reason why the authors chose to let the action representation be independent of the state?
>
> The learned action representation does not strictly need to be state-independent. Indeed, we have explored the possibility of allowing the action representation to be dependent on the latent state. Through empirical testing on the Quadruped Run and Walker Run tasks (**Figure 15 (Right)** of the attached PDF), we find that whether the action representation is state-dependent or independent does not create a major difference in the performance of TACO. Thus, we opted for the simpler approach, allowing the action representation to be state-independent. We acknowledge that our empirical results do not rule out the possibility of allowing state dependent action representations, and we will clarify this in our revised manuscript.
>
> >Question 4: Is the action representation being compressed?
>
> The action representation in our approach is not primarily aimed at compression. Instead, our focus is on shaping this representation to align with the optimal Q function. Drawing an analogy with state representation, we assert that a valuable action representation ought to be able to linearly represent the optimal Q function and should also be predictive of subsequent states. These guiding principles inform our method in TACO, where the action embedding is systematically learned through a thoughtful integration of critic TD loss and temporal contrastive loss updates. We appreciate the reviewer's insight and question, and we will clarify this point in our revised manuscript.

---

> > ### Comment · Reviewer_gh9U · 2023-08-15
> >
> > Thank you for you response; the additional experiments and clarifications address my concerns, and I am now comfortable raising my score from Weak Accept to Accept.
> >
> > The results in this paper are useful to the community, as they underscore the benefits of learning both state and action representations -- a concept that is fairly unexplored in the literature -- and the empirical results are strong in the online setting. Since performance differences in the offline setting are smaller, it would be easier to interpret results if you highlighted cells for which e.g. the difference between CQL/TD3+BC w. TACO and CQL/TD3+BC is statistically significant according to a paired t-test at a 95% confidence level. The significance is obvious for some tasks (e.g. CQL in Quadruped Run, Full-replay) but not so obvious for others.
> >
> > A few additional comments:
> > 1. To clarify the experiments in lines 215-231 I suggest replacing "Then we select four actions...from a standard normal distribution" in lines 223-227 with what you wrote in your rebuttal: "we sample four actions, $a_1, a_2, a_3, a_4 \in \mathbb R^6$, to act as centroids. For each of the four centroids, we generate 1000 augmented actions by adding standard Gaussian noises to the last 20 dimensions." I now see how that particular phrase was the source of my confusion.
> > 2. I don't think you specify what the shaded regions denote in Fig. 4.

---

> > > ### Author Response · Authors · 2023-08-15
> > > **Thank you!**
> > >
> > > Thank you for your response and additional comments. We are encouraged by your recognition of our work's importance in state-action representation learning for visual continuous control and our algorithm's strong empirical performance. We will integrate all of your suggestions, including highlighting tasks where our method significantly outperforms offline RL baseline algorithms, into our final manuscript.

---

### Author Rebuttal · Authors · 2023-08-08

We thank all reviewers for their insightful questions and valuable feedback. We are encouraged that reviewers recognize the importance of our tackled problem in state-action representation learning (gh9U). They also appreciate the flexibility and applicability of our proposed approach, TACO, in both offline and online RL settings (1TK9, Y2ee, od4S), and highlight the outstanding empirical performance of our methods (gh9U, 1TK9, Y2ee, od4S). We have addressed all individual questions of reviewers in separate responses.

Additionally, we have attached a one-page PDF with more experimental results, including (**Figure 15**) two ablation studies - one substituting CURL with SODA, another using state-dependent over state-independent action representation; (**Figure 16**) an additional ablation study on CURL and reward prediction loss in offline RL; (**Figure 17**) online RL experiments for six Meta-world manipulation tasks; and (**Figure 18**) further experiments on representation pretraining using TACO.

Here we briefly outline the response to the two most common questions raised by the reviewers. The detailed response is included in each individual response.

**CURL and reward prediction objectives** (Reviewer gh9U, Y2ee):
In our initial tests on the Quadruped Run and Walker Run tasks, we discovered that CURL and reward prediction enhanced TACO's performance, leading us to incorporate these two losses into TACO's final objectives. However, we would like to emphasize again that the CURL and reward prediction objective are added only as auxiliary losses to further improve the performance. We conduct additional experiments on both online (Reviewer Y2ee) and offline RL (Reviewer gh9U) settings. Together with **Figure 6(b)** in our original manuscript, they demonstrate that CURL and reward prediction objectives are not the main driver of the superior performance of TACO.

**Novelty of our approach compared with other representation learning methods such as DRIML** (Reviewer 1TK9, oD4S):
We want to emphasize other representation learning objectives, except for CURL and ATC, were studied in environments with well-represented, small discrete action spaces, thus overlooking the importance of action representation learning. In contrast, we identify the importance of action representation learning in continuous control, an under-explored topic in previous works. We introduce TACO as a simple yet effective approach to utilize temporal contrastive loss to learn state and action representation. In our comparison with other representation learning objectives in the **Table 2** of our original manuscript, we fix the action representation learning part and focus only on the design of temporal contrastive loss. Still, we see some non-trivial improvements over other representation learning objectives. In summary, our contribution lies in both underscoring action representation in continuous control tasks and introducing a simple yet effective temporal contrastive TACO loss as a solution to state and action representation learning in visual continuous control problems.

---

### Author Response · Authors · 2023-08-15
**Additional Questions?**

Thank you all for the constructive and insightful feedbacks. We have conducted additional experiments and provided clarifications to address all the questions that you raised, as we believe that they are crucial for enhancing our paper’s quality. If you have any additional questions or concerns, we are more than happy to address them.

---

### Decision · Program_Chairs · 2023-09-21

**Decision:**

Accept (poster)

**Comment:**

Reviewers are unanimous in recommending acceptance (although to varying degrees). I have gone through the discussions in detail and am glad to see that these discussions have resulted in an improved paper. I do not think there are any significant concerns remaining and recommend acceptance.